# OpaqueToolsBench: Learning Nuances of Tool Behavior Through Interaction

## Abstract

Tool-calling is essential for Large Language Model (LLM) agents to complete real-world tasks. While most existing benchmarks assume simple, perfectly documented tools, real-world tools (e.g., general "search" APIs) are often *opaque*, lacking clear best practices or failure modes. Can LLM agents improve their performance in environments with opaque tools by interacting and subsequently improving documentation? To study this, we create OpaqueToolsBench, a benchmark consisting of three distinct task-oriented environments: general function calling, interactive chess playing, and long-trajectory agentic search. Each environment provides underspecified tools that models must learn to use effectively to complete the task. Results on OpaqueToolsBench suggest existing methods for automatically documenting tools are expensive and unreliable when tools are opaque. To address this, we propose a simple framework, ToolObserver, that iteratively refines tool documentation by observing execution feedback from tool-calling trajectories. Our approach outperforms existing methods on OpaqueToolsBench across datasets, even in relatively hard settings. Furthermore, for test-time tool exploration settings, our method is also efficient, consuming 3.5-7.5$\times$ fewer total tokens than the best baseline. [1]

## 1 Introduction

Tools expand the knowledge and capabilities of Large Language Model (LLM) agents beyond their learned parameters (Schick et al., 2023). With the right tools, LLMs can search the web, send emails, execute code, and interact with the world. Yet this extension fails when tools lack adequate documentation – a pervasive problem in deployed systems. Real-world APIs are complicated to explain, enterprise functions lack specifications, and domain-specific tools ship with minimal descriptions. Moreover, some tools are difficult (or impossible) to fully explain, even for humans, e.g., often the optimal usage of search engines or LLM-based QA APIs is not known, even to their creators. Such tools are **opaque**: their behavior unpredictable, their correct usage unknown. As shown in Figure 1, tool opacity harms the performance of LLM agents, a problem that compounds when models must coordinate multiple tools for complex tasks.

This raises a fundamental question: **Can LLM agents learn to use opaque tools by observing their behavior during interaction?** Such capability would transform tool-calling agents from brittle systems dependent on perfect specifications into adaptive ones that can improve through experience.

We introduce OpaqueToolsBench, a benchmark for learning in opaque tool settings where tool documention is underspecified. Different than existing benchmarks that assume near-perfect tool specifications like ToolBench (Qin et al., 2023), APIBench (Patil et al., 2023), and Berkeley Function Calling Leaderboard (Patil et al., 2025), OpaqueToolsBench does *not* provide comprehensive function signatures, detailed descriptions, or explicit type specifications. OpaqueToolsBench spans three distinct environments: general function calling, interactive chess playing, and long-trajectory agentic search. Each provides underspecified tools that models must learn to use effectively, testing both single-instance discovery and cross-instance learning. Unlike existing benchmarks that focus on measuring model capacity to compose concrete, well documented tools, OpaqueToolsBench is designed to measure an LLM agent's ability to adapt to the imperfect documentation of tools through interaction.

---

[1] We release our code, data, and benchmark at `anonymous.com`

**Query**: *How long am I contagious with the flu?*

Figure 1: LLM agents may struggle when presented with *opaque tools* – tools lacking clear description of their usage best practices or their failure modes. To succeed in these settings, we posit that LLM agents must explore tool usage to learn their true behaviors.

We evaluate current approaches that optimize tool descriptions, including Play2Prompt (Fang et al., 2025) and EasyTool (Yuan et al., 2024). Both methods fall short on OPAQUETOOLSBENCH: systems either focus on compression of existing tool documentation, completely neglecting the interaction with tools (EasyTool) or require single-tool exploration phases separate from task execution, which becomes expensive in some settings (Play2Prompt). As a result, these methods are ineffective and sometimes expensive, often consuming thousands of tokens in preliminary exploration before attempting the composite task.

We propose an alternative framework, TOOLOBSERVER, that refines tool documentation by observing and learning from execution feedback acquired through composite task trajectories. On OPAQUETOOLSBENCH, our method exceeds baseline performance consistently by on average 18.6%, while consuming 3.5-7.5× fewer tokens in test-time settings. Our results demonstrate that learning from execution feedback provides an efficient path to handling opaque tools in real-world environments. It also demonstrates that LLM agents can adapt to underspecified tools through interaction, making tool-calling viable even in poorly-documented environments.

## 2 BACKGROUND

Language models estimate the conditional distribution $P(x_t|x_{<t})$ over a vocabulary $\mathcal{V}$, where $x_{<t}$ represents all preceding tokens (Radford et al., 2019; Brown et al., 2020). Despite their impressive results on language benchmarks, language models nonetheless face fundamental limitations: their knowledge is frozen, they can't take actions in the real world, and they are often unreliable on simple capabilities like adding numbers. These constraints have motivated the development of *tool-augmented* language models that invoke external functions to overcome these limitations (Schick et al., 2023; Mialon et al., 2023).

**Tool-Calling in Language Models** Language models interact with tools through a structured interface that enables them to extend their capabilities beyond parametric knowledge. Each tool $t_i$ in the available tool library $\mathcal{T} = \{t_1, t_2, \ldots, t_n\}$ is characterized by:

$$t_i = \{n_i, d_i, p_i, e_i\}$$

where $n_i$ is the name of the function, $d_i$ is the documentation of the function behavior, $p_i$ is an optional dictionary of parameters consisting of their name, description, and whether or not they are required. $e_i$ is the executable function itself. In practice, parameter information from $p_i$ is often incorporated into the behavioral documentation $d_i$ as a string description.

Given a user query $q$, the language model $M$ generates tool calls through a systematic process of **retrieving** then **calling**, all through autoregressive decoding. The model conditions on both the query and the available tool descriptions in their context to produce a sequence that may include tool invocations:

$$s \sim P_M(s|q, \mathcal{T}) = P_M(s|q, (n_1, d_1, p_1), \ldots, (n_n, d_n, p_n)) \tag{1}$$

where $s$ is the sampled sequence. When the model determines a tool is needed, it produces a structured tool call $c = \langle n_i, \text{args}_i \rangle$ as part of this sequence, where $\text{args}_i$ must conform to the parameter specification $p_i$. Upon generating a tool call, the system executes $r_i = e_i(\text{args}_i)$ and appends the result to the context. The model then continues its autoregressive generation, now conditioning on the expanded context:

$$s' \sim P_M(s'|q, \mathcal{T}, c, r_i) \tag{2}$$

This process may repeat, with the model invoking multiple tools or generating a final response that incorporates the tool outputs to address the user's query. Such tool calling language models are often used to complete goal-oriented tasks and are referred to as LLM agents (Wang et al., 2023b).

## 3 THE OPAQUETOOLSBENCH BENCHMARK

Prior work creating and evaluating LLM agents assumes well-defined and well-documented tools (Guo et al., 2024a; Qin et al., 2023; Shen et al., 2024; Patil et al., 2023). However, many real-world tools are black boxes whose behavior can only be understood through interaction. As an example, consider an agent given access to a set of off-the-shelf Search APIs provided as tools. Such APIs may index documents at differing granularities and covering different domains/times/topics/styles, may (or may not) be keyword based, might automatically run multiple hops, and could even themselves use language models to orchestrate the search process. For LLM agents to perform optimally in such an environment, they need to learn these nuances by *interacting with these tools* and *observing the feedback*. Furthermore, the proliferation of Model Context Protocols (MCP) [2] have made it easy to connect LLM agents to tools that come with variable quality and accuracy of tool documentation.

We characterize such tools as **opaque**. Reflecting on the diverse challenges described above, from the inherent complexity of Search APIs to the variable documentation quality of MCP tools, we find that opacity stems from two distinct sources. We distinguish between two types of opacity – both practically important – that motivate our benchmark design:

- **Type 1: Documentation Opacity** Tools whose behavior is deterministic and describable, but whose documentation is inaccurate or missing. This is often inevitable in real-world settings: legacy systems often rely on unwritten "tribal" knowledge, while open marketplaces like the Model Context Protocol (MCP) contain inconsistent descriptions at scale.

- **Type 2: Intrinsic Opacity** Tools that are opaque due to inherent complexity (e.g., search engines, LLM-based tools, complex simulations). These tools often have a simple schema but complex, undocumented behavioral nuances. For example, a search API's ranking logic is hard to fully capture a priori; even the tool creator cannot predict how a neural system handles every edge case. The agent must instead learn these nuances through interaction.

To effectively operate in environments characterized by these forms of opacity, LLM agents need to demonstrate the following abilities:

1. **Manipulate structured & natural language inputs:** Certain tools that expose traditional REST APIs (for example, currency conversion) have structured inputs. Whereas others like Search APIs accept a string where the nuances are encoded in natural language, which are more open-ended/opaque.

2. **Learn from process feedback:** Opaque tools can sometimes only be understood by observing a trajectory/sequence of tool uses, rather than just the result from a single call, e.g., you might not be able to discern if a search API's output is "good" until you try to compose the result with other information.

3. **Learn across trajectories:** In many real-world settings with a fixed set of tools, LLM agents need to accumulate experience from prior trajectories, e.g., using a search tool with one goal in mind should help one gain experience useful for using that tool for a different goal.

4. **Test-time generalization:** In settings where new tools are available at test time, we need to test the ability of LLM agents to learn their nuances while completing the task itself.

---

[2] modelcontextprotocol.io

Table 1: An overview of the datasets in OPAQUETOOLSBENCH

|  | BFCL-Opaque | Chess | BrowseComp Domains |
|---|---|---|---|
| **Description** | Complete user's request by correctly calling a function with a limited description. | Play chess by choosing one of many specialized chess engine tools every turn. | Compose domain-specific search tools to locate hard-to-find information. |
| **Skills required** | Structured inputs, process feedback, test-time generalization | Process feedback, learn across trajectories, tool sequencing | Unstructured inputs, process feedback, learn across trajectories, tool sequencing |
| **Settings** | **Documentation quality**
1. Anon. function names
2. Anon. function names + description
3. Anon. function names + parameter names | **Tool sets (chess engines)**
1. Beginner, intermediate, advanced skill
2. Opening, midgame, endgame, late-endgame specialists | **Tool sets (search tools)**
1. Domain-specific (9)
2. Domain-specific (9) + Full Search |
| **Evaluation** | Evaluation Accuracy, Param Acc., AST Acc. | % of Optimal Tool Calls, ELO | Accuracy, # Tool Calls |
| **# Train / # Test** | - / 90 | 200 / 1800 | 83 / 747 |

5. **Learn tool sequencing:** Certain tool behaviors may only manifest in conjunction with other specific tool calls. For example, if tool B can only be called after a successful invocation of tool A, the agent needs to be able to create such trajectories to learn nuances of tool B.

We introduce OPAQUETOOLSBENCH to systematically evaluate these abilities across both types of opacity. It consists of three environments: an opacified version of BFCL (Patil et al., 2023) we call BFCL-Opaque (targeting **Type 1**), a novel game-playing environment based on Chess (targeting **Type 2**), and BrowseComp Plus with opaque search (Chen et al., 2025) (targeting **Type 2**). Our environments are designed to test the above aspects of learning the behavior of opaque tools while being reproducible and efficient to run. A summary of our datasets and key information is shown in Table 1 and in the following paragraphs. Examples from the tasks are shown in Figure 2.

**BFCL-Opaque:** The Berkeley Function Calling Leaderboard (BFCL) (Patil et al., 2025) consists of question-functions-answer tuples with functions from multiple programming languages and diverse application domains. Following Fang et al., 2025, we use the executable subset so feedback is available from executing the tools. The tasks are simple, for example, fetching the current weather or computing the area of a polygon. Each task has between 2-4 tools and on average 3. We modify this environment to evaluate **test-time generalization** and **Type 1 Opacity** – though the tools of each instance may slightly overlap a few others instances', we instead treat the tools of each test instance as independent and unknown. Specifically, we modify this environment by obfuscating the function names as well as completely removing all docstrings. We do this independently for each problem – so the same function across different test instances will be named differently. For each test instance, the agent now has to learn what the function does and produce structured output in the form of its arguments and their types. We create progressively easier settings by providing either (1) only function description or (2) only parameter names. We use a binary task completion rate as our evaluation metric. Refer to Appendix C.1 for details on environment construction.

**Chess** In this environment the task is to win against a fixed-strength chessbot (Stockfish (Romstad et al., 2024) at search depth 2; higher=better). Instead of playing moves directly (as in Zhang et al. (2025)), models are given access to a set of move suggesting tools. At each turn, the LLM agent uses one of several undocumented tools that accept current board positions in FEN notation and play the move that's suggested. While the tool interfaces are identical, we introduce undocumented behaviors in each tool. Specifically, in the first setting, each tool uses Stockfish but with different search depths (2, 4 & 8), testing if the agent can learn fine-grained discrimination between tools based on the final outome. In the second setting, we provide tools optimized for opening, middlegame, endgame, and late endgame, enabling us to test the ability to explore and learn temporal patterns. Since the same set of tools are used across different instances, we test agent's ability to learn tool behavior across trajectories (capturing **Type 2 Opacity** where behavior depends on game state). Refer to Appendix C.2 for details on environment construction.

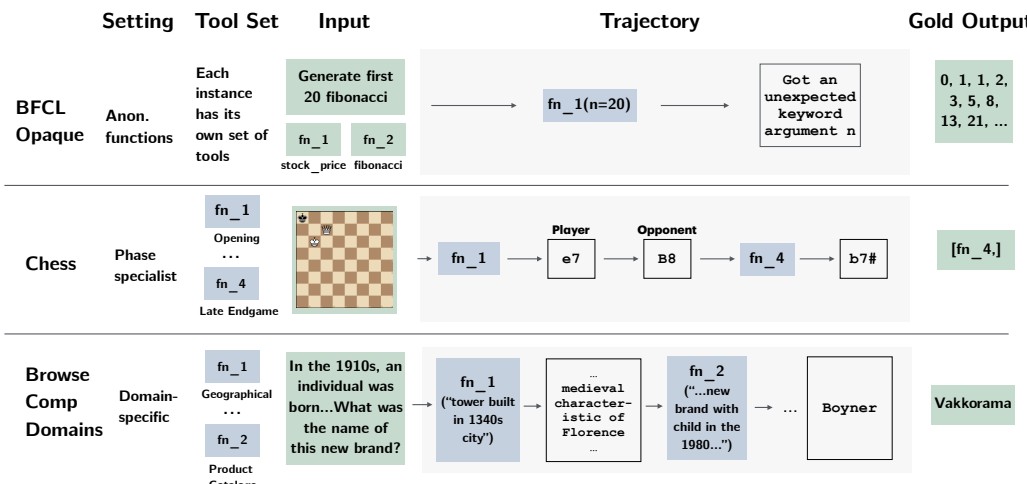

Figure 2: Examples from each of the environments in OPAQUETOOLSBENCH. For Chess and BrowseComp Domains, each setting has a fixed set of tools across all instances (chess and search engines respectively). In BFCL-Opaque, each instance has custom query-dependent tools. The dataset defines the input and output (green). For each instance, the agent makes tool calls iteratively (blue). For BFCL-Opaque and BrowseComp Domains we check for a match with the gold answer. For Chess, we match engine choices in green with optimal engine choices.

**BrowseComp Domains** The BrowseComp dataset consists of question and short answer pairs which measure the ability of agents to locate hard-to-find, entangled information on the internet, and might require browsing tens or even hundreds of websites in the process. BrowseComp Plus (Chen et al., 2025) further improves it by fixing the retrieval corpus and thus making this benchmark easy to reproduce. We leverage this corpus and create anonymous domain-specific search tools (4-6 tools) by partitioning the corpus (academic papers, product catalogs, geographical data, news articles). The LLM agent must not only discover the domain specialization of each tool, it must also learn to create optimal search queries for each tool and learn to search in the right sequence (addressing **Type 2 Opacity** in query formulation). Each instance in our dataset has a corresponding sequence of optimal tool calls. We compare the tool sequence with the optimal sequence and use accuracy as our evaluation metric. Refer to Appendix C.4 for details on environment construction.

These three environments test for the four key abilities of LLM agents: (1) With BFCL-Opaque we test for structured inputs and with BrowseComp Domains unstructured inputs, (2) All environments allow learning from intermediate outputs of tool calls, (3) BrowseComp Domains and Chess test if the LLM agent can learn across trajectories, while BFCL-Opaque checks for generalization to new tools at test time, (4) Chess and BrowseComp Domains test for LLM agent ability to sequence opaque tools in the right sequence.

## 4 TOOLOBSERVER: IMPROVING DOCUMENTATION VIA INTERACTION

Existing approaches for optimizing tool documentation have limitations that render them less effective in complex tool-calling environments. For example, EasyTool (Yuan et al., 2024) relies primarily on compressing existing long documentation of tools into more concise instructions, making it unsuitable to settings where the tool documentation is underspecified or completely lacking. Meanwhile, Play2Prompt (Fang et al., 2025) adopts an iterative refinement strategy, but it requires an isolated single-tool exploration phase separate from task execution, making it inefficient and (as we will show) not performant in complex environments that require a long trajectory of tool calls.

To address these limitations, we propose TOOLOBSERVER. TOOLOBSERVER requires no up-front documentation, and discovers and refines tool documentation through observation and reflection on execution trajectories. The key idea is to alternate between an *Exploration Phase* and a *Reflection Phase*. During exploration (line 3 of Algorithm 1), we collect execution trajectories using the current documentation $\mathcal{D}^{(k-1)}$, which can initially be null. There a separate reasoning model analyzes the

trajectories, identifies patterns, and updates tool descriptions accordingly. We repeat this process iteratively, improving tool descriptions and exploring better trajectories in the next iteration.

OPAQUETOOLSBENCH contains two settings: (1) with **shared tools** between train and test set and (2) with **unseen tools** that can only be observed at test time (and there is no train set). We instantiate TOOLOBSERVER in two modes to accommodate the two settings: **offline mode**, which pre-optimizes documentation during training when all test instances share a common tool set and **online mode**, which optimizes documentation at test time for previously unseen tools.

## 4.1 OFFLINE MODE

In offline mode, we pre-optimize documentation using a set of training instances sharing the same tools. Algorithm 1 outlines the procedure. The core intuition is to establish a feedback loop where the agent attempts tasks, observes execution outcomes against ground truth, and iteratively refines its understanding of tool behaviors from this signal. This approach leverages two key advantages of the offline setting: (1) multiple training instances and (2) the ability to evaluate the final answer. We discuss the two key phases below:

---

**Algorithm 1** TOOLOBSERVER: Offline mode

**Require:** Initial descriptions $\mathcal{D}_0$, iterations $K$, LLM Agent $M_A$, Editor LLM $M_E$
**Ensure:** Optimized documentation $\mathcal{D}^*$
1: Initialize $\mathcal{D}^{(0)} \leftarrow \mathcal{D}_0$
2: **for** $k = 1$ to $K$ **do**
3:      // **Exploration Phase**
4:      $\mathcal{T}_k \leftarrow \text{CollectTrajectories}(\mathcal{D}^{(k-1)}, M_A)$
5:      // **Reflection Phase**
6:      $\mathcal{D}^{(k)} \leftarrow \text{ReflectAndUpdate}(\mathcal{T}_k, \mathcal{D}^{(k-1)}, M_E)$
7: **end for**
8: **return** $\mathcal{D}^{(K)}$

---

**Exploration Phase: Collecting Trajectories** The goal of this phase is to generate diverse interaction traces that reveal the latent behaviors and failure modes of the opaque tools. To do this, we execute the agent $M_A$ on the training set $\mathcal{X}_{train}$ using the current tool documentation $\mathcal{D}^{(k-1)}$. By running across the entire training set, we ensure the agent explores tool behaviors across a wide variety of contexts (e.g., diverse chess positions or search queries) and diverse usage patterns. We use temperature sampling to collect a breadth of trajectories $\mathcal{T}_k = \{\tau_1, ..., \tau_{|\mathcal{X}_{train}|}\}$, where each trajectory $\tau_i$ captures the full sequential decision process: starting from the initial state, it records the alternating sequence of reasoning steps, tool calls, and environmental observations (e.g., tool execution outputs) leading to the final outcome.

**Reflection Phase: Reflecting and Updating Documentation** The goal of this phase is to distill raw interaction experience into explicit, generalizable tool documentation. However, processing the full set of trajectories $\mathcal{T}_k$ simultaneously is infeasible due to context window constraints and the difficulty of extracting consistent patterns from massive, noisy data streams. To address this, we employ a meta-prompting strategy with an **"Editor" LLM** $M_E$ (this can be the same as our LLM agent $M_A$). It analyzes trajectories via a hierarchical process:

1. **Batch Analysis (with ground truth):** We first split the trajectories into mini-batches to manage context limits. For each batch, we explicitly task the Editor with updating the tool descriptions based on the execution history. Acting as a local reasoner, it identifies causal links between tool usage patterns and success/failure outcomes. Crucially, it is provided with the **ground truth** (e.g., the gold answer) or a **task performance signal** derived from the training environment. Using this signal, it distinguishes effective usage from failures and generates a candidate description specific to that batch. Consequently, this phase produces a diverse set of local descriptions, where each captures insights valid for its specific subset of trajectories.

2. **Consensus Merge:** Since observations from a single batch may be noisy or overfit to specific instances, a second Editor pass aggregates the candidate proposals from all batches. It acts as a **consensus filter**, retaining only those behavioral rules that appear consistently across multiple diverse batches while discarding instance-specific hallucinations.

This two-stage process is critical for the offline mode: it allows us to scale to large datasets while ensuring the learned documentation generalizes across different contexts. We repeat this exploration and reflection loop for $K$ iterations. Finally, we run the LLM agent with the optimized documentation $\mathcal{D}^{(K)}$ on the test set. Further implementation details, including prompt templates and dataset-specific details, can be found in Appendix A and E.

## 4.2 ONLINE MODE

In the online mode, we are only given access to a single test instance with a set of unseen tools. For a single test instance with unique tools $\mathcal{T}_x$ that cannot be experimented with apriori, we first collect a trajectory by executing the agent's generated tool calls and recording the real-time environment feedback (e.g., return values or error messages). Then, we use the **Editor LLM** to analyze that trajectory and update the tool documentation based on the observed behavior(s). We repeat this process for a maximum of $K$ iterations. If the Editor does not update the documentation, we stop the iteration early. Note that in online mode the gold output is not used as part of the process. We use the final version of the tool documentation to run the LLM agent on the test instance and compute an evaluation metric using the gold output.

## 4.3 COMPARISON TO PLAY2PROMPT

While Play2Prompt (Fang et al., 2025) also adopts an iterative refinement strategy, our method differs from it in several crucial ways. First, Play2Prompt requires initial documentation $D_0$ to bootstrap their reverse generation process; they first generate valid tool invocations based on this documentation, then construct matching queries. In contrast, our method can operate under the more challenging setting where tools are completely opaque, requiring discovery purely through task-driven exploration. Second, Play2Prompt also requires optimization of all tools individually beforehand. In contrast, our method explores all tools at once, balancing an exploration/exploitation tradeoff and tool interactions vs. being forced to exhaustively explore all tools individually apriori. In §5, TOOLOBSERVER's design leads to notably better performance under opaque tool settings.

## 5 EXPERIMENTS

We benchmark three contemporary tool-calling LLM agents on OPAQUETOOLSBENCH. First, we test on GPT-5 (OpenAI, 2025), the most capable model OpenAI model for reasoning and agentic tool use at the time of writing. We also use GPT-5-mini, a cost-efficient yet still capable version. We use the ReAct framework (Yao et al., 2022) to iteratively reason and call functions. Further details can be found in Appendix A.

**TOOLOBSERVER and Baselines Experimental Details** For our main experiments with TOOLOB-SERVER, we also use GPT-5 as our model for reflecting on and updating tool descriptions (editor model). F ollowing TOOLOBSERVER, for all baselines we use GPT-5. We include (1) **Play2Prompt** (Fang et al., 2025), which improves tool-documentation from self-play followed by self-reflection and (2) **EasyTool** (Yuan et al., 2024) which automatically rewrite the tool documentation by condensing tool descriptions and creating structured functional guidelines.

### 5.1 MAIN RESULTS

Our main results in Tables 2-4 demonstrate both the challenge of existing baselines on OPAQUE-TOOLSBENCH, and the strong performance of TOOLOBSERVER, showing its potential to improve LLM agent performance in opaque tool settings.

**TOOLOBSERVER outperforms baselines on BFCL-Opaque and recovers near-optimal performance.** Table 2 demonstrates that TOOLOBSERVER outperforms all baselines across documentation levels and LLM agents. In the function name only setting, TOOLOBSERVER recovers 0.62 execution accuracy with GPT-5, whereas Play2Prompt only achieves 0.50. Furthermore, because Play2Prompt exhaustively test hundreds of tools, most of which will not be relevant, the total input + output tokens consumed for exploration is $\sim$1.7M, which is $7.5\times$ more than the exploration budget of TOOLOBSERVER. Across documentation levels, this ranges from $3.5\times$ - $7.5\times$. This indicates that for test-time optimization settings, our method is both effective and relatively inexpensive.

We identify an initially challenging but tractable scenario: when parameter information is missing. OpenAI agents initially obtain 0 performance, repeatedly calling tools without arguments. However, TOOLOBSERVER effectively bridges this gap. By comparing against a Gold Oracle (perfect documentation), we find our method recovers 93% of the performance gap in the underspecified setting (0.86 vs. 0.92). To verify this is due to valid schema discovery rather than luck, we an-

Table 2: Performance on BFCL across models and baselines. We use ReAct. Gold is the ground truth documentation and base is the opacified set. TO denotes TOOLOBSERVER, P2P denotes Play2Prompt, and ET denotes EasyTool. Columns denote **E** (Execution-based overall accuracy), **P** (Parameter accuracy), and **A** (AST accuracy). The highest **E** value in each row is **bolded**.

| Documentation | Model | ReAct | | | | | | | | | | | | | | |
| | | Gold | | | Base | | | + TO | | | + P2P | | | + ET | | |
| | | E | P | A | E | P | A | E | P | A | E | P | A | E | P | A |
| Tool Setting: Individual Tools per Problem | | | | | | | | | | | | | | | | |
| Anon. Fn. Names Only | GPT-5 | 0.92 | 0.91 | 0.93 | 0 | 0 | 0.59 | **0.62** | 0.61 | 0.83 | 0.50 | 0.54 | 0.80 | 0 | 0 | 0.59 |
| | GPT-5-mini | 0.94 | 0.90 | 0.95 | 0 | 0 | 0.59 | **0.62** | 0.54 | 0.82 | 0.52 | 0.54 | 0.82 | 0 | 0 | 0.59 |
| Anon. Fn. + Real Desc. | GPT-5 | 0.92 | 0.91 | 0.93 | 0 | 0 | 0.59 | **0.86** | 0.82 | 0.89 | 0.50 | 0.47 | 0.80 | 0 | 0 | 0.60 |
| | GPT-5-mini | 0.94 | 0.90 | 0.95 | 0 | 0.01 | 0.60 | **0.80** | 0.77 | 0.90 | 0.46 | 0.45 | 0.79 | 0 | 0 | 0.60 |
| Anon. Fn. + Param Names | GPT-5 | 0.92 | 0.91 | 0.93 | 0.78 | 0.82 | 0.93 | **0.82** | 0.85 | 0.94 | 0.78 | 0.80 | 0.94 | 0.70 | 0.71 | 0.90 |
| | GPT-5-mini | 0.94 | 0.90 | 0.95 | 0.82 | 0.85 | 0.94 | **0.88** | 0.88 | 0.96 | 0.84 | 0.84 | 0.95 | 0.74 | 0.76 | 0.93 |

Table 3: Average percentage of best tool calls (Acc) and ELO rating at a given state for Chess. TO denotes TOOLOBSERVER, P2P denotes Play2Prompt, and ET denotes EasyTool. Gold is the ground truth documentation. The highest value in each row is **bolded**.

| Model | ReAct | | | | | | | | | |
| | Gold | | Base | | + TO | | + P2P | | + ET | |
| | Acc | ELO | Acc | ELO | Acc | ELO | Acc | ELO | Acc | ELO |
| Tool Setting: Opening, midgame, endgame, late-endgame specialists | | | | | | | | | | |
| GPT-5 | 64.4 | 1411 | 23.5 | 728 | **40.1** | 1020 | 35.8 | 966 | 23.2 | 761 |
| GPT-5-mini | 52.8 | 1243 | 24.9 | 772 | **32.1** | 949 | 19.5 | 754 | 25.8 | 739 |
| Tool Setting: Beginner, intermediate, advanced skill | | | | | | | | | | |
| GPT-5 | 100 | 2341 | 24.9 | 1572 | **29.1** | 1756 | 0.25 | 1622 | 24.9 | 1584 |
| GPT-5-mini | 100 | 2346 | 25.7 | 1674 | **28.4** | 1778 | 22.7 | 1481 | 25.5 | 1645 |

alyze Parameter (P) and AST (A) accuracy. TOOLOBSERVER consistently outperforms baselines on these granular metrics (e.g., 0.61 Parameter Accuracy vs. 0.54 for Play2Prompt), confirming it successfully synthesizes correct input structures from scratch.

**BrowseComp Domains and Chess are challenging, but TOOLOBSERVER can still discover tool nuances** As shown in Table 3 and 4, BrowseComp Domains and Chess are empirically much harder tasks than BFCL Opaque – for Chess, the best performance for any model, measured by percentage of best tool calls, is only 40.1%. On the other hand, the best accuracy for BrowseComp Domains is only 24.1%. In both of these cases, despite the initial difficulty of the tasks, our method TOOLOBSERVER outperforms baselines across tool settings in both domains, achieving the best performance.

However, even the optimized chess performance is seemingly low - with only 29.1 on the skill tool setting with only three tools with GPT-5. We posit that this is because our evaluation metric is strict – during exploration, the model observes tool usage from three different tools, all of which will perform well against the opponent. When evaluating via *best tool %* , they are penalized even if they choose a relatively good tool. In contrast, the Streaming ELO scores – computed from win-rates against diverse opponents – verify that the learned documentation enables practically effective gameplay, improving over the baseline even if the strictly optimal tool is not always chosen.

For BrowseCompPlus, comparisons against the Gold Oracle (33.1% accuracy for GPT-5) reveal that this task is intrinsically difficult even with perfect documentation. TOOLOBSERVER recovers some of this gap while also reducing the number of average tool calls. This intuitively makes sense: as documentation is improved for search tools, the LLM agent is able to route queries more effectively, both improving downstream performance and reducing wasted search calls. Overall, the success of TOOLOBSERVER across both offline settings demonstrate the strength of our method: reflecting on and adapting based on real trajectories improves performance better than isolated tool testing.

Table 4: Performance comparison table on BrowseComp Domains using Qwen-0.6B. Acc. refers to accuracy and #TC refers to number of tool calls. TO denotes TOOLOBSERVER, P2P denotes Play2Prompt, and ET denotes EasyTool. The highest Acc. value in each row is **bolded**.

| Model | ReAct | | | | | | | | | |
| | Gold | | Base | | + TO | | + P2P | | + ET | |
| | Acc. | #TC | Acc. | #TC | Acc. | #TC | Acc. | #TC | Acc. | #TC |
| --- | --- | --- | --- | --- | --- | --- | --- | --- | --- | --- |
| Tool Setting: Domain-specific (9) Search | | | | | | | | | | |
| GPT-5 | 30.8 | 22.6 | 18.8 | 25.3 | **20.3** | 23.2 | 19.4 | 23.8 | 18.7 | 23.8 |
| GPT-5-mini | 28.9 | 21.8 | 14.6 | 23.2 | **18.8** | 22.9 | 15.1 | 12.1 | 15.0 | 24.4 |
| Tool Setting: Domain-specific (9) + Full Search | | | | | | | | | | |
| GPT-5 | 33.1 | 20.5 | 21.4 | 24.8 | **24.1** | 21.9 | 23.8 | 14.4 | 20.6 | 25.5 |
| GPT-5-mini | 30.9 | 22.2 | 20.3 | 23.3 | **22.1** | 21.0 | 21.0 | 18.6 | 20.8 | 23.7 |

## 5.2 ANALYSIS

We analyze a few components of TOOLOBSERVER and our own OPAQUE-TOOLSBENCH:

**TOOLOBSERVER performance over iterations** Figure 3 shows that when minimal tool information is available (plot A), the model gradually improves across iterations, measured by execution accuracy on BFCL-Opaque. When some useful information is already provided (plot B), performance spikes early on then plateaus, with little additional gain. Finally, in plot C, where most of the important tool is available, improvements are negligible and performance saturates almost immediately. These patterns are consistent across all evaluated models. However, GPT-5 takes more iterations to converge at the final tool documentation, while GPT-5-Mini consistently converges sooner.

We also report the average reflection iterations required for convergence in Table 5.

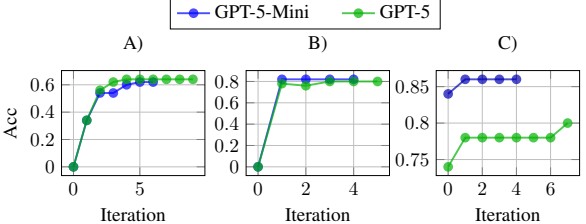

Figure 3: Performance of TOOLOBSERVER on BFCL-Opaque with increased iterations on the tool settings: A): Anon. Function Names, B): Anon. Function Names + Descriptions, C): Anon. Function Names + Param. Names. Iterations stop after full convergence. These are expanded versions of the Table 2 results.

Table 5: Average number of reflection iters. required by TOOLOBSERVER to converge on BFCL-Opaque.

| Configuration | GPT-5 | GPT-5-mini |
| --- | --- | --- |
| A) Anon. Fn. Names | 3.44 | 2.96 |
| B) Anon. Fn. Names + Desc. | 2.66 | 2.60 |
| C) Anon. Fn. Names + Param. | 2.22 | 2.24 |

We observe two key trends. First, convergence speed correlates with information availability: as starting documentation improves (from anonymized names to including parameters), average iterations decrease by $\approx 35\%$ (from 3.2 to 2.2). Second, we observe that in the most opaque setting, GPT-5 uses more iterations (3.44) than GPT-5-mini (2.96). Combined with the performance gap in Figure 3, this suggests the stronger model engages in more thorough exploration to uncover complex behaviors. Even in the hardest setting, convergence occurs in under 3.5 iterations on average, demonstrating the experience-efficiency of TOOLOBSERVER.

**Fidelity of leaarned descriptions** We assess the quality of generated documentation both quantitatively and qualitatively. First, we quantify the fidelity of learned documentation by measuring semantic (SBERT embedding Reimers & Gurevych (2019)) and lexical (ROUGE-1; Lin (2004)) similarity against gold standards on BFCL-Opaque (Table 6). TOOLOBSERVER consistently outperforms baselines on both metrics. In the hardest "Anon. Fn. Names" setting, we achieve a semantic similarity of **0.58** (vs. 0.51 for Play2Prompt) and a lexical overlap of **0.28** (vs. 0.18). The improvement in both metrics confirms that TOOLOBSERVER captures both the general semantic meaning *and* specific terminology and functional constraints from the gold documentation.

Second, we qualitatively analyze the descriptions generated for Chess tools in Appendix B. Interestingly, TOOLOBSERVER learns the nuances of *when* each tool must be used. For example, the middle game specialist final description explicitly mentions the tool is useful for stabilizing dynamic middle

Table 6: Average similarity metrics (semantic and lexical) of final documentation generated vs. gold documentation on BFCL-Opaque. The highest sem. is bolded, while the highest lex. is italicized. TO denotes TOOLOBSERVER, P2P denotes Play2Prompt, and ET denotes EasyTool.

| Documentation | TO (GPT-5) | | TO (GPT-5-Mini) | | P2P | | ET | |
|---|---|---|---|---|---|---|---|---|
| | Sem. | Lex. | Sem. | Lex. | Sem. | Lex. | Sem. | Lex. |
| Anon. Fn. Names | 0.44 | 0.21 | **0.58** | *0.28* | 0.51 | 0.18 | 0 | 0 |
| Anon. Fn. Names + Desc. | **0.78** | 0.43 | 0.78 | *0.44* | 0.70 | 0.31 | 0.71 | 0.43 |
| Anon. Fn. Names + Param. Names | **0.71** | *0.40* | 0.71 | *0.40* | 0.69 | 0.28 | 0.69 | 0.39 |

games. A similar pattern is observed in the depth specialist: the depth-2 specialist is described as effective at spotting immediate conversions, while the depth-16 specialist is ideal for volatile situations. On the other hand, Play2Prompt doesn't perform well in the depth specialization setting – with a shocking performance of 0.25 – since the best tool's description is under-specified. Instead, the agent always picks the *second* best tool which has a well-specified tool description.

**Strength of LLM Agent vs Editor LLM** We ablate the editor, using weaker models, i.e. GPT-5-Mini and O3, and measure the performance on BFCL in Table 7. The strength of the editor model directly affects the performance of both agents. Furthermore, the stronger agent model is generally more robust to a weak editor

Table 7: Performance on BFCL-Opaque with different editors on Anon. Fn. Names setting

| LLM Agent | GPT5 | GPT5-mini | o3 |
|---|---|---|---|
| GPT5 | 0.64 | 0.62 | 0.60 |
| GPT5-mini | 0.62 | 0.54 | 0.48 |

model. GPT-5-Mini sees a steep drop of 8 points when using GPT-mini as the editor. However, with GPT-5 as the LLM agent and GPT-5-Mini as the editor, the observed drop is a meager 2 points.

## 6 RELATED WORK

**LLM agents** LLM Agents that can call tools enable interaction with external systems. Schick et al. (2023) showed language models can learn to use tools through self-supervised learning, while ReAct (Yao et al., 2022) introduced the reasoning-action paradigm where agents alternate between thinking and acting. Shinn et al. (2023) further enhanced tool-using agents with the ability to learn from failures through verbal self-reflection. These advances have enabled complex agent behaviors, from multi-agent coordination (Park et al., 2023) to continuous skill acquisition (Wang et al., 2023a).

**Tool Documentation** Hsieh et al. (2023) demonstrate the critical role of comprehensive documentation in tool learning. Recent work has focused on automatically refining these descriptions: Qu et al. (2024) introduces the DRAFT framework where gathered experience is used to rewrite tool documentation, while Fang et al. (2025) introduce Play2Prompt, which employs self-play followed by self-reflection to iteratively improve tool documentation. Yuan et al. (2024) propose EASYTOOL, showing that condensing verbose descriptions and adding usage examples significantly improves downstream performance by reducing hallucination rates. Wang et al. (2024) extend this line of work by incorporating short-term and long-term memory mechanisms after self-reflection phases.

**Tool Evaluation Benchmarks** Current tool-use benchmarks include BFCL (Berkeley Function Calling Leaderboard), which evaluates simple, single-turn function calls, ToolBench (Xu et al., 2023) and StableToolBench (Guo et al., 2024b), which offer thousands of real-world REST APIs. However, ToolBench and StableToolBench suffer from API key accessibility issues and poor reproducibility due to their reliance on external services.

## 7 CONCLUSION

We introduce OPAQUETOOLSBENCH, a benchmark consisting of three goal-oriented environments with opaque tools, which better reflects the reality of working with underspecified, poorly documented tools in real-world scenarios. Through our proposed TOOLOBSERVER framework, we demonstrate that LLM agents can effectively learn to improve tool documentation through iterative observation of execution feedback, achieving superior performance while being significantly more token-efficient than existing approaches.

## LLM USAGE STATEMENT

We declare that we have used LLMs for polishing the content of this paper.

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

## A  EXPERIMENTAL DETAILS

We list further experimental details and results in this section.

### A.1  TOOLOBSERVER HYPERPARAMETERS

For all datasets, we use the ReAct (Yao et al., 2022) framework, which prompts the agent to generate interleaved reasoning traces and executable tool calls to solve tasks dynamically.

**Max iterations**: For BFCL Opaque, we use a max iterations of 10 to iteratively test and optimize tool descriptions. For BrowseComp and Chess, we use a max iterations of 3 due to computational constraints (these trajectories are much larger and we have many more of them). Recall that for BFCL, the LLM agent can stop early if it reaches a final answer; we report the average number of iterations taken in Table 2.

**Offline Batch Size** For the two tasks which we use the offline version of TOOLOBSERVER for - Chess and BrowseComp Domains – we use a mini-batch size of 10 trajectories.

**Chess Specifics:** To ensure computational efficiency during the *exploration phase*, we simulate partial trajectories of 10 moves starting from the sampled position. We evaluate the state at the end of this partial trajectory using the board value estimate from Stockfish (Romstad et al., 2024). This scalar value acts as the dense process reward described in §4.1, allowing the editor language model to critique the strategic quality of tool choices without needing to reach a checkmate.

**Generation Details** Our experiments are primarily done with GPT-5 and GPT-5-mini [3]; we sample from these models (OpenAI does not allow setting temperature or top-p parameters for these models). To reduce latency during exploration on Chess which has long trajectories, we set reasoning effort to be minimal. Everywhere else, we use "medium" reasoning effort, including on the documentation updates.

Finally, in Figures 6 - 12 in Appendix E, we detail the prompts we use across our three datasets for 1) tool usage/exploration 2) documentation updates and 3) the prompts for merging the learned descriptions (if applicable).

### A.2  OPAQUETOOLSBENCH DATASET DETAILS

**BrowseComp Domains** We run BrowseComp Domains using the Qwen-0.6B embedder model, following Chen et al., 2025. Also following this work, we set the max number of tokens in retrieved snippets to be 512, and the number of retrieved documents per search tool to be $k = 5$.

**Chess** For the *final evaluation* on the test set (Table 3), we simulate full games. We enforce a maximum limit of **120 moves**; if the game exceeds this limit without a decisive result (checkmate/stalemate), it is recorded as a draw.

## B  COMPARISON OF FUNCTION DESCRIPTIONS FOR CHESS TOOLS

### B.1  GROUND TRUTH

The following are the ground truth descriptions generated for different chess tools.

> **Opening Specialist**
>
> A chess player that specializes in the opening phase. Plays strong moves (depth 16 analysis) during the opening, but resorts to random moves during middlegame and endgame phases. Uses game phase detection to determine current phase.

---

[3]https://platform.openai.com/docs/models/gpt-5 and https://platform.openai.com/docs/models/gpt-5-mini

**Middle Game Specialist**

A chess player that specializes in the middlegame phase. Plays strong moves (depth 16 analysis) during the middlegame, but resorts to random moves during opening and endgame phases. Uses game phase detection to determine current phase.

**End Game Specialist**

A chess player that specializes in the endgame phase. Plays strong moves (depth 16 analysis) during the endgame, but resorts to random moves during opening, middlegame, and late endgame phases. Uses game phase detection to determine current phase.

**Late Game Specialist**

A chess player that specializes in the late endgame phase. Plays strong moves (depth 16 analysis) during the late endgame, but resorts to random moves during opening, middlegame, and regular endgame phases. Uses game phase detection to determine current phase.

**Depth-2 Specialist**

Analyzes the current chess position using Stockfish engine at depth 2 (shallow analysis). Takes a FEN string as input and returns the best move in algebraic notation. This provides quick but less accurate move suggestions.

**Depth-8 Specialist**

Analyzes the current chess position using Stockfish engine at depth 8 (moderate analysis). Takes a FEN string as input and returns the best move in algebraic notation. This provides a good balance between speed and accuracy.

**Depth-16 Specialist**

Analyzes the current chess position using Stockfish engine at depth 16 (deep analysis). Takes a FEN string as input and returns the best move in algebraic notation. This provides highly accurate move suggestions but takes more time.

## B.2 TOOLOBSERVER

The following are the descriptions generated by our method for different chess tools.

**Opening Specialist**

Best as a consolidation tool when already worse or in simplified positions, especially endgames. It favors quiet, low-variance moves that limit further damage. Less suitable for seizing the initiative or equalizing from balanced positions, and it can misjudge in sharp middlegames or make superficial material grabs in quiet endings. Use it to steady the ship, not to press for precise gains.

**Middle Game Specialist**

Best for stabilizing dynamic middlegames and defusing pressure through sound structural choices, particularly when defending. In calm or technical endgames and balanced positions, it tends toward passivity or inaccurate king/pawn decisions and is not a precise converter. Choose it to neutralize threats, not to fine-tune technical outcomes.

**End Game Specialist**

Best in simplified, technical positions that reward precision – especially rook endgames – and when an accurate, active move is needed to maintain control from an equal or slightly worse stance. Avoid it in sharp, tactical middlegames or when low-risk, resilient defense is required; it can overreach and swing the evaluation heavily. It's also not ideal for clinging on in lost endgames as White.

**Late Game Specialist**

Best for holding calm, technical endgames and maximizing resistance with safe, non-committal moves. Serviceable as a middle option when the position isn't tactically charged. Avoid it in sharp or tactical situations and whenever king safety is delicate; it tends to overlook immediate threats and can trigger large single-step collapses.

**Depth-2 Specialist**

Best when you want a decisive, forcing solution. Excels at spotting immediate conversions – tactical captures, central pawn breaks, direct mating or queening threats – and at jump-starting counterplay through rapid coordination (e.g., quick castling, active piece placement). Ideal for positions where concrete calculation can resolve tension right away. Tends to be high-variance: strong at seizing chances, but can overpress or misjudge tactical safety in messy defenses. Less suited to slow consolidation or risk-minimizing play.

**Depth-8 Specialist**

Best for building and sustaining initiative with forcing play. Excels at creating and maintaining pressure through checks, rook lifts, and energetic pawn breaks, and will alter the structure when it strengthens activity. Strong at converting an edge by keeping the opponent on the back foot. Less reliable when the position demands quiet consolidation or a concrete defensive neutralization; can overlook the need to stabilize before pressing.

**Depth-16 Specialist**

Best for stabilization and neutralization. Excels at consolidating king safety, coordinating pieces, recapturing accurately, and simplifying when under pressure. Finds direct defensive resources – exchanges and file contests – that reduce the opponent's practical chances. Ideal in worse or volatile positions and whenever risk control is paramount. Tends to forgo sharp attacking continuations in favor of solid, resilient play.

## B.3 PLAY2PROMPT

The following are the descriptions generated by Play2Prompt for different chess tools.

## Opening Specialist

function_1 analyzes a chess position from a FEN string and returns a single best-move suggestion in SAN (Standard Algebraic Notation).

**Parameters**

- `board_state` (string, required): Complete 6-field FEN: `<pieces> <active> <castling> <en-passant> <halfmove> <fullmove>`. Provide a legal position and include all fields, separated by single spaces. Example: `rnbqkbnr/pppppppp/8/8/8/8/PPPPPPPP/RNBQKBNR w KQkq - 0 1`

**Output**

- A SAN move string, e.g., "e4", "Nf3", "exd5", "O-O", "O-O-O", "Bb5+", "Qh7#".

**Usage example**

- Call name: `function_1` (do not use the original name).
- Input JSON:
  `{ "board_state": "rnbqkbnr/pppppppp/8/8/8/8/PPPPPPPP/RNBQKBNR w KQkq - 0 1" }`
- Example result: "e4"

**Notes**

- Use the exact key `board_state`.
- Output is SAN only (not UCI/LAN).

## Middle Game Specialist

Analyzes a standard chess position and returns a single best move suggestion.

**Input:**
A complete 6-field FEN string.

**Parameter:**

- `board_state` (string, required) — Valid FEN with:
  1. piece placement (8 ranks separated by "/", digits for empty squares, pieces PNBRQK/pnbrqk),
  2. active color "w" or "b",
  3. castling rights as a subset of "KQkq" or "-",
  4. en-passant target square "-" or a3–h6,
  5. halfmove clock (non-negative integer),
  6. fullmove number ($\geq$1).

**Output:**
One move in Standard Algebraic Notation (SAN), e.g., "e4", "Nf3", "exd5", "Qh5+", "e8=Q#", including disambiguation as needed. No extra text.

**Scope:**
Standard chess only (not Chess960).

**Validation:**
Malformed/illegal FEN may be rejected — ensure correct field count and values.

**Examples:**

- `board_state: "rnbqkbnr/pppppppp/8/8/4P3/8/PPPP1PPP/RNBQKBNR b KQkq - 0 1"` $\rightarrow$ "d5"
- `board_state: "r1bqkbnr/pppppppp/2n5/8/8/2N5/PPPPPPPP/R1BQKBNR w KQkq - 0 3"` $\rightarrow$ "Nf3"

**End Game Specialist**

Analyzes a chess position and returns a single move suggestion in Standard Algebraic Notation (SAN). Alias: `endgame_specialist`. Input must be a complete, valid FEN string (six space-separated fields): piece placement, side to move (w/b), castling rights, en passant target square, halfmove clock, fullmove number. The side to move is taken from the FEN.

**Parameters:**

- `board_state` (string, required): Full FEN for the current position. Include correct castling rights and en passant target if applicable.

**Output:**

- One SAN move string (not UCI/coordinate), e.g., "Nf3", "exd5", "O-O", "O-O-O", "a8=Q", "Qh7+", "Rxf8#", with standard disambiguation as needed. Captures use "x"; promotions use "=Q/R/B/N"; checks "+"; checkmates "#".

**Example:**

- `board_state:` `"rnbqkbnr/pppppppp/8/8/8/8/PPPPPPPP/RNBQKBNR w KQkq - 0 1"` → "e4"

**Late Game Specialist**

Analyzes a chess position and returns a single move suggestion in Standard Algebraic Notation (SAN) as a plain string.

**Parameters**

- `board_state` (string, required): Full 6-field FEN of the current position. Format: "¡piece placement¿ ¡side to move¿ ¡castling rights¿ ¡en passant target¿ ¡halfmove clock¿ ¡fullmove number¿". Example: `"rnbqkbnr/pppppppp/8/8/8/8/PPPPPPPP/RNBQKBNR w KQkq - 0 1"`. Must represent a legal position.

**Output**

- SAN move string only (no JSON object). Examples: "e4", "Nf3", "exd5", "O-O", "O-O-O", "e8=Q", "Qh7#", "Rd1+".

**Notes**

- Input must be FEN (not PGN or UCI).
- SAN uses uppercase piece letters (pawn omitted), "x" for captures, "+/#" for check/mate, and "=Q/R/B/N" for promotions.

**Example call**

- `function_4({ "board_state": "rnbqkbnr/pppppppp/8/8/8/8/PPPPPPPP/RNBQKBNR w KQkq - 0 1" })`

**Example response**

- "e4"

**Depth-2 Specialist**

Suggests a single chess move for the side to move in the given position using a shallow fixed-depth analysis (~2 plies). Always call this tool when a move is requested from a FEN; do not infer moves without it.

**Parameters:**

- `board_state` (string, required): A valid full FEN for standard chess with all 6 fields: piece placement, active color (w/b), castling rights (KQkq or -), en passant target (square or -), halfmove clock, fullmove number. The FEN must be legal and consistent; castling and en passant fields affect legality.

**Output:**
A single move in Standard Algebraic Notation (SAN), not UCI/LAN. Examples: "e4", "Nf3", "O-O", "R1e2", "exd5", "e8=Q#". Includes "+" or "#" if applicable. No extra text.

**Notes:**
If multiple moves are near-equal, one is returned.

**Example:**
`board_state="rnbqkbnr/pppppppp/8/8/8/8/PPPPPPPP/RNBQKBNR w KQkq - 0 1"` → "e4".

**Depth-8 Specialist**

Analyzes a standard chess position and returns the single best move at fixed search depth 8 plies. Use exactly one required parameter.

**Required parameter:**

- `board_state` (string): A single-line valid FEN (Forsyth–Edwards Notation). Must include exactly 6 space-separated fields:

    1. piece placement,
    2. side to move (w/b),
    3. castling rights (KQkq or -),
    4. en passant target square (e.g., e3 or -),
    5. halfmove clock (integer),
    6. fullmove number (integer).

    No extra whitespace or newlines; standard chess only; position should be legal.

**Returns:**
One move in SAN (Standard Algebraic Notation), e.g., "Nf3", "exd5", "O-O", "O-O-O", "e8=Q", "Rxd8+", "Qh7#" (with disambiguation if needed). Not UCI; no scores or move lists.

**Example call:**
`{"board_state":"rnbqkbnr/pppppppp/8/8/4P3/8/PPPP1PPP/RNBQKBNR b KQkq - 0 1"}`

**Example output format:**
"Nf6"

> **Depth-16 Specialist**
>
> Analyzes a standard chess position and returns the engine's best move as a single SAN string.
>
> **Parameter:**
>
> - `board_state` (string, required) — a Forsyth–Edwards Notation (FEN) of a legal position; include side-to-move and other fields (full 6-field FEN preferred). The move is determined from the FEN's side-to-move.
>
> **Output:**
> Exactly one move in Standard Algebraic Notation (SAN), e.g., "e4", "Nf3", "O-O", "exd5", "Qh8#", "b8=Q+", "R1e2". Do not expect UCI/LAN or any extra text.
>
> **Notes:**
> Fixed internal search depth (cannot be configured); standard chess only (no variants). Use the exact key name `board_state` and pass only the FEN string.
>
> **Example call:**
> `{"board_state":"rnbqkbnr/pppppppp/8/8/4P3/8/PPPP1PPP/RNBQKBNR b KQkq - 0 1"}`
>
> **Example response:**
> "d5"

B.4 EASYTOOL

The following is the description generated by EasyTool for chess tools; note that they are almost all identical since the initial description is the same (differences are only up to sample differences).

> **All tools**
>
> **Tool Description:** 'function_1' analyzes a chess position from a FEN 'board_state' and returns a suggested move in algebraic notation.
>
> **Tool Guidelines:**
> *Scenario:*
> if you want to analyze a chess position (e.g., the Ruy Lopez after 4...Nf6) and get a suggested move for White.
> *Parameters:*
> `{"board_state":"r1bqkb1r/1ppp1ppp/p1n2n2/4p3/B3P3/5N2/PPPP1PPP/RNBQK2R w KQkq - 2 5"}`

# C Tasks

## C.1 BFCL-Opaque: Discovering Tool Functionality from Opaque Descriptions

The Berkeley Function Calling Leaderboard (BFCL) (Patil et al., 2025) We systematically degrade BFCL's tool descriptions to create an opaque setting where models must infer functionality through interaction. This tests the fundamental ability to map between ambiguous tool interfaces and their underlying behaviors.

**Task Setup:** Models receive user queries requiring specific tool calls (e.g., "What's the weather in San Francisco?", "Schedule a meeting for tomorrow at 3pm") but must discover which tools accomplish each task. We provide tools with systematically degraded documentation: function names are replaced with generic identifiers (e.g., `tool_1`, `tool_2`), descriptions are removed or made ambiguous, and parameter specifications lack type information or semantic hints. Models must experiment with different tools and parameter combinations to discover correct usage patterns.

**Tool Degradation Strategy:** Initially, the complete tool specifications include function descriptions, parameter names, and parameter descriptions. We replace semantic function names with generic identifiers (`function_1`, `function_2`, etc.), removing the primary semantic cue for tool selection. We then create three different documentation levels:

1. **Anon. function name only**, where we remove everything (function description, parameter names, and parameter descriptions), testing pure behavioral discovery through trial and error with **no** documentation

2. **Anon. function name + Description**, where we remove all parameter names/descriptions while keeping only the function description, testing whether models can infer argument structure from behavioral descriptions alone.

3. **Anon. function name + Parameter names**, where we remove the function description and parameter descriptions while keeping only parameter names, testing discovery of functionality from argument structure without semantic guidance.

**Data Collection:** We evaluate on BFCL's executable subset, which provides deterministic, programmatically verifiable tasks across four categories: executable simple, executable multiple function, executable parallel function, and executable parallel multiple function. This subset ensures reproducible evaluation—each task has ground-truth tool calls and expected outputs.

**Primary Evaluation Metrics:** We measure binary task completion accuracy—whether the model successfully calls the correct tool with proper arguments to satisfy the user query.

**Enhanced Evaluation Metrics** Standard BFCL evaluation relies on a binary success metric (Pass/Fail). To better diagnose *how* agents learn opaque tool behaviors, we introduce two granular metrics that distinguish between semantic understanding (selecting the right tool) and syntactic mastery (calling it correctly).

**1. Parameter Accuracy.** This metric measures the exact correctness of the arguments provided, conditional on the agent selecting the correct tool. If the model chooses the wrong function, the score is 0. When the correct function is chosen, we calculate the percentage of expected parameters that are perfectly recovered. Specifically, it is the ratio of arguments where both the parameter name and the assigned value exactly match the ground truth, divided by the total number of required parameters. This metric strictly penalizes missing arguments or incorrect values, distinguishing agents that "know" the tool from those that merely guess the function name.

**2. AST (Abstract Syntax Tree) Accuracy.** AST Accuracy evaluates the structural validity and "grammar" of the tool call, independent of whether the values are correct. It is calculated as the average of five components:

- **Format Validity:** Whether the output is parsable as valid JSON or a Python Abstract Syntax Tree (using `ast.parse`).

- **Structure Validity:** Whether the parsed object contains the standard `function` and `args` keys.

- **Type Correctness:** The percentage of parameters where the data type (e.g., string, integer, list) matches the ground truth schema.

- **Schema Compliance:** A strict check ensuring the structure is valid, all types are correct, and no hallucinated parameters exist.

- **Hallucination Check:** Whether the agent generated parameters that do not exist in the tool definition.

## C.2 CHESS: LEARNING STRATEGIC TOOL SELECTION THROUGH EXPERIENCE

We challenge LLMs to play Chess, but instead of predicting moves directly, models are given access to several undocumented tools that accept current board positions in FEN notation and return move recommendations. Each next move suggestion function has an identical interface, but undocumented behavioral differences. Thus, the agent must discover through gameplay that each implements different strategies. Performance directly reflects the model's ability to document each tool's strengths.

**Task Setup:** Models play chess games against a fixed-strength opponent (Stockfish at depth 2) by selecting from available tool sets. Each trajectory consists of a max number of moves where the model must select a tool and play against the fixed opponent.

**Tool Sets:** We construct two tool sets of increasing complexity:

1. **Phase specialization** (4 tools) - These tools are engines that work well for specific phases: opening, middlegame, endgame and late endgame. These phases are defined by number of pieces on the board: opening phase has at least 28 pieces, middlegame at least 16, endgame at least 10 and late endgame has less than 10 pieces. Each tool plays moves according to a strong engine (depth 16 analysis) in its own phase but plays randomly otherwise. An optimal agent would learn to document these temporal patterns

2. **Depth gradients** (3 tools) Tools 1, 2, and 3 are Stockfish with search depth 2, 4, and 8 (higher=better); this tests fine-grained discrimination between similar high-quality tools.

**Data Collection:** We sample 2000 chess positions from the Lichess database[4], which provides hundreds of millions of positions with chess engine evaluations. We split these 2000 positions into training (10%) and test (90%) sets, maintaining the same stratified distribution across both game phases and position evaluations to ensure comparable evaluation conditions:

- **Game phase** (determined by piece count): opening (25%), middlegame (40%), endgame (25%), late endgame (10%)

- **Position evaluation** (from Lichess engine analysis): equal positions (40%), slight advantages for white/black (10% each), winning positions for white/black (8% each), and crushing/mate positions for each side (6% each)

This stratified sampling ensures models encounter diverse board states that test tool selection across different game scenarios.

**Main Evaluation Metrics:** Since we know the optimal tool call at every turn (the correct phase specialized tool for phase specialization or the highest search depth tool for depth gradients), we simply calculate the accuracy of LLM agent tool calls as our evaluation metric.

## C.3 ADDITIONAL CHESS EVALUATION METRIC: STREAMING ELO

To provide a fine-grained measurement of strategic decision quality beyond binary tool-choice accuracy, we implement a **Streaming Elo** rating system. The Elo rating system is a method for calculating the relative skill levels of players in zero-sum games.

---

[4] https://database.lichess.org

**Opponent Pool.**    We evaluate the agents against a diverse set of deterministic opponents using the Stockfish engine at varying difficulty levels to represent different tiers of play:

- **Beginner:** Stockfish Level 1 (Approx. Elo 800)

- **Intermediate:** Stockfish Level 5 (Approx. Elo 1600)

- **Master:** Stockfish Level 10 (Approx. Elo 2400)

**Update Rule.**    The agent starts with a standard baseline rating of $R_0 = 1200$. After each game $i$, the rating is updated based on the result against an opponent with rating $R_{opp}$. We use a K-factor of $K = 32$. The expected score $E_i$ and the updated rating $R_{i+1}$ are calculated as follows:

$$E_i = \frac{1}{1 + 10^{(R_{opp} - R_i)/400}} \tag{3}$$

$$R_{i+1} = R_i + K \cdot (S_{actual} - E_i) \tag{4}$$

where $S_{actual}$ is the game outcome (1.0 for a win, 0.5 for a draw, 0.0 for a loss).

**Experimental constraints and Bootstrapping.**    Due to the high computational cost of running full tool-augmented chess trajectories, we evaluate on a subset of 300 games played against the opponent pool. To prevent infinite loops in drawn or lost positions, any game exceeding **120 moves** is automatically adjudicated as a draw.

Streaming Elo ratings can be sensitive to the specific chronological order of matches (e.g., facing a string of Master-level opponents early can depress the rating, making recovery difficult). To eliminate this variance and ensure a robust final metric, we employ **bootstrapping**. We shuffle the sequence of the 300 completed games into **1,000 random permutations**, calculate the final streaming Elo for each permutation, and report the mean rating across all permutations.

### C.4   BROWSECOMP DOMAINS: LEARNING MULTI-TOOL COORDINATION FOR COMPLEX INFORMATION SEEKING

Complex question-answering requires discovering not just individual tool capabilities, but how to coordinate multiple tools strategically. BrowseComp Plus (Chen et al., 2025) provides an ideal testbed for this challenge—human-curated questions that demand synthesizing information from dozens of search queries. Unlike simple retrieval tasks, these questions require models to discover through interaction which tools access which information sources, how to formulate effective queries for each, and how to combine results to build comprehensive answers.

**Task Setup:**    Models must answer complex, multi-hop questions using search tools with opaque documentation. Each question requires aggregating information from multiple sources—for example, comparing statistics across countries, tracing historical developments, or synthesizing technical specifications. While BrowseComp Plus provides a fixed corpus containing all necessary documents, models receive no documentation about which tools search which subsets or how query syntax varies between tools. They must discover these constraints through experimentation during actual question-answering trajectories.

**Tool Sets and Degradation Strategy:**    We construct two search environments that test different aspects of tool discovery: (1) **Domain-specific search** (9 tools) where specialized tools each query distinct document subsets (academic papers, product catalogs, geographical data, news articles), testing discovery of tool coverage boundaries and domain-specific query patterns ; and (2) **Mixed search** (10 tools) which combines specialized domain tools with a general tool that searches the entire corpus, testing strategic selection between targeted and broad search approaches. We introduce realistic opacity patterns that mirror production search systems—tools are provided with generic names (`search_1`, `search_2`) and minimal documentation. Models must discover through interaction: coverage boundaries (which document types each tool can access), query constraints (maximum query length, required syntax, boolean operator support), and ranking behaviors (how each

tool prioritizes results by recency, relevance, or popularity). These undocumented behaviors only emerge through varied usage patterns across multiple queries.

**Data Collection:** We use BrowseComp Plus's curated question set, which includes 830 complex questions designed to require extensive information gathering. Questions span diverse domains including science, history, geography, and current events. Each question has human-validated answers and requires on average 15-30 search queries when using well-documented tools, making this an ideal benchmark for measuring if models can learn tool capabilities while solving real tasks.

**Evaluation Metrics:** We measure both answer accuracy (F1 score against gold answers) and search efficiency (number of queries required). Unlike single-shot benchmarks, we track improvement across questions—does the model become more efficient at using discovered tool capabilities? We also measure cross-question transfer: when models discover a tool searches academic papers while answering a science question, can they apply this knowledge to a history question requiring scholarly sources?

# D  BASELINES

Following TOOLOBSERVER, for all baselines we use GPT-5.

## D.1  PLAY2PROMPT

**Play2Prompt** (Fang et al., 2025) improves tool-documentation from self-play followed by self-reflection. It iteratively generates a set of tool usage examples by "playing" with the tool, using the responses until it generates valid example tool usages. Using these examples, the documentation is iteratively improved based on the tool use errors observed while using the current documentation.

## D.2  EASYTOOL

(Yuan et al., 2024) which automatically rewrite the tool documentation in two stages. First, it condenses the tool descriptions to eliminate redundant information and focuses only on core functionality. Then, it creates structured functional guidelines with usage scenarios and parameter examples to help LLMs understand when and how to use each tool. EasyTool is limited by its lack of execution of the tools themselves. Furthermore, the descriptions and functional guidelines are beforehand, hence cannot benefit from any knowledge gained as the trajectory rolls out.

# E  TOOLOBSERVER PROMPTS

```
You are an expert in composing and exploring functions. You are given a
user question and a set of available tools.

You must call at least one tool in response to every user question.
There are no exceptions. Refusing to call a tool is not allowed.

If you are confident in a tool's purpose, use it appropriately to
address the user's request. If you are unsure what a tool does, make a
best guess and try it with plausible parameters to learn how it behaves.
 It is better to experiment than to fail to respond.

Always format tool calls correctly with all required parameters. You
should only return function calls in the tool call sections.
```

Figure 4: BFCL Exploration Prompt

```
"pre": """You are improving function documentation by analyzing real
usage examples. Your goal is to write descriptions that help future
users understand exactly what each function does and how to use it
correctly.

Current Function Definitions:
{available_functions}

Observed usage examples:
""",
--------------------------------------------------------------------------

"middle": """
Example {example_num}:
User Question: {question}
Function Called: {function_call}
Function Output: {function_output}
""",
--------------------------------------------------------------------------

"post": """
## Important

**If the existing description is already accurate, do not include it in
your response.** Only provide updates for functions that actually need
improvement. You can tell a description is accurate if in the usage
examples the function was called successfully and produced the expected
output.

## Instructions

**Only analyze and update functions that were actually called in the
examples above.** Do not provide descriptions for functions that weren't
 used.

For each function that was called:
1. Examine what inputs were provided and what the function returned
2. Compare this with the existing description
3. **Only provide updated descriptions for functions that need
improvement**

## Requirements for Updated Descriptions

Each improved description must clearly explain:
- **What the function does** (its purpose and behavior)
- **What inputs it expects** (parameter types, format, requirements)
- **What it returns** (output format and content)
- **How to call the function** (usage pattern or syntax)

Write 1-2 clear, comprehensive sentences that give users everything they
 need to use the function correctly.

## Format

FUNCTION: [function_name]
DESCRIPTION: [Your improved description here]

## Updated Descriptions (only for functions that need improvement):"""
```

Figure 5: The BFCL reflection prompt. It is split into three parts. A "pre-prompt", a "middle-prompt", and a "post-prompt". We concatenate them together along with the real usage behaviors (ie, the function calls and the outputs).

```
You are a chess-playing AI assistant with access to chess analysis tools
 for evaluating positions and suggesting moves.

<objective>
Play the strongest move in the current position by making exactly ONE
tool call to analyze the position.
</objective>

<tool_selection_strategy>
- Make exactly ONE tool call per move decision
- Choose the strongest tool available for the current position
- If tools appear similar in capability or are similarly unknown,
consider trying a different tool than recently used ones to diversify
your information sources
- When one tool is clearly superior for the position type, use it
regardless of history
</tool_selection_strategy>

<exploration_principle>
- Primary goal: Select the strongest tool for each position
- Secondary consideration: If multiple tools seem equally strong or
equally unknown, vary your selection based on recent usage history
- This diversification helps avoid potential blind spots from relying on
 a single tool's perspective
- Never sacrifice move quality for exploration - only explore when tools
 are genuinely comparable
</exploration_principle>

<decision_framework>
With your single tool call, consider:
- What type of position is this? (tactical, positional, endgame, opening
)
- Which tool is strongest for this specific position?
- If multiple tools seem equally strong, which have I used recently?
- Is there a clear best tool, or are several tools comparably suitable?
</decision_framework>

<tool_preamble>
Before making your tool call:
- Explain which tool you're selecting and why it's the strongest choice
for this position
- If multiple tools seemed equally viable, briefly note why you selected
 this one over the others
</tool_preamble>

<quality_checks>
- Select the strongest available tool (or make a reasonable choice among
 equals)
- Make exactly one tool call
</quality_checks>
```

Figure 6: Chess exploration prompt. We append the board state and the recent tool-calls to this prompt before receiving the next tool-call.

```
Analyze chess tool performance across N game trajectories to generate
improved tool descriptions that clearly differentiate when to use each
tool.

<input>
- Game trajectories with tool calls, moves, and positions
- Board evaluations (positive=White advantage, negative=Black advantage)
- Current tool descriptions
- Side played by agent in each game
</input>

<analysis_requirements>
For each tool:
- Identify consistent patterns in its behavior and performance
- Determine what distinguishes it from other tools
- Provide concrete proof: cite specific trajectories and moves showing
these patterns
- Focus on situations where this tool performs differently than others

Evaluation notes:
- Higher eval is better for White, lower eval is better for Black
- IMPORTANT: Always compare tools relatively, not absolutely
- Example for White: Tool A suggesting move to +2 is better than Tool B
suggesting +1
- Example for Black: Tool A suggesting move to -3 is better than Tool B
suggesting -1
- Critical: Even in losing positions, compare which tool finds the best
continuation
  * For White: -5 is much better than -10 (both losing, but one is more
resilient)
  * For Black: +10 is much better than +15 (both losing, but one offers
more resistance)
- Don't dismiss a tool just because it suggested moves in bad positions
- focus on whether it found the BEST move among the alternatives
</analysis_requirements>

<output_per_tool>
**Tool: [name]**

Observed patterns: [Key behaviors identified with specific trajectory
evidence]

Distinguishing characteristics: [What makes this tool different from
others, with examples]

Updated description:
[Concise description stating when to use this tool relative to others]

Reasoning: [Justification based on trajectory evidence]
</output_per_tool>

<final_output>
After analyzing all tools, provide a decision framework for selecting
between tools based on the patterns discovered.
</final_output>

Key: Every claim must reference trajectories. Descriptions must be
comparative (tool X better than Y for Z) not absolute.
```

Figure 7: The Chess batch analysis reflection prompt. The chess trajectories are appended to this prompt.

```
You will receive N LLM responses, each analyzing different batches of
chess game trajectories. Synthesize these into definitive tool
descriptions.

<synthesis_task>
For each tool:
1. Identify patterns that appear across multiple responses
2. Note contradictions between responses
3. Distinguish true patterns from batch-specific noise
4. Look for emergent patterns that no single analysis identified but
become visible when viewing all analyses together
5. Create ONE final description based on the most reliable patterns

Critical:
- A behavior mentioned in only 1-2 responses is likely batch-specific
noise
- Focus on patterns that multiple independent analyses discovered
- Also identify meta-patterns: behaviors that emerge from the collective
 evidence but weren't explicitly stated in any single response
- When responses conflict, examine their evidence strength
- Final descriptions should capture the tool's strengths/weaknesses but
NOT explicitly name other tools
</synthesis_task>

<output_format>
**Tool: [name]**

Synthesis reasoning:
[Explain which patterns were most consistent across analyses, what
emergent patterns were discovered, how conflicts were resolved, and why
certain behaviors were included/excluded in the final description.]

Final description:
[Single definitive description of when to use this tool. Describe its
characteristics and optimal use cases WITHOUT referencing other tools by
 name. Example: "Best for tactical positions requiring deep calculation.
 Excels at finding forcing sequences and material sacrifices. Tends to
be overly aggressive in quiet positions."]
</output_format>
```

Figure 8: The Chess consensus merge reflection prompt. The chess descriptions generated from the previous step are appnded to this prompt.

```
You are a question-answering AI assistant with access to search tools
that return different types of results.

<objective>
Find the correct answer to the question by making strategic tool calls
over multiple turns. Each turn, you make exactly ONE tool call and
receive its results before deciding your next action.
</objective>

<how_this_works>
- You will be called multiple times for the same question
- Each time, you'll see the full history of your previous tool calls and
 their results
- Each turn, make exactly ONE tool call to gather more information
- Use what you've learned from previous tool calls to inform your next
choice
- Once you have enough information, provide your final answer
</how_this_works>

<tool_selection_strategy>
- Make exactly ONE tool call per turn
- Choose the strongest tool available for the current information needs
- Review what you've already learned from previous tool calls
- If tools appear similar in capability or you're uncertain about what
they return, consider trying a different tool than recently used ones to
 diversify your information sources
- When one tool is clearly superior for the remaining information needs,
 use it regardless of history
</tool_selection_strategy>

<exploration_principle>
- Primary goal: Select the strongest tool for your current information
gap
- Secondary consideration: If multiple tools seem equally strong or you'
re uncertain about their outputs, vary your selection based on what you'
ve already tried
- This diversification helps avoid potential blind spots from relying on
 a single tool's perspective
- Never sacrifice answer quality for exploration - only explore when
tools are genuinely comparable or unknown
- Learn from previous tool results: if a tool gave poor results before,
consider alternatives
</exploration_principle>

<decision_framework>
Each turn, consider:
- What information do I still need to answer this question?
- What have I learned from previous tool calls?
- What type of question is this? (factual, current events, historical,
technical, domain-specific)
- Which tool is strongest for filling my current information gap?
- If multiple tools seem equally strong or I'm uncertain about them,
which have I used already?
- Do I have enough information to answer, or should I make another tool
call?
</decision_framework>
```

Figure 9: BrowseComp Domains exploration prompt, part 1. We append the context, including the recent tool-calls and results, to this prompt before receiving the next tool-call.

```
<tool_preamble>
Before making your tool call:
- Review what you've learned from previous tool calls (if any)
- Explain which tool you're selecting and why it's the strongest choice
for your current information needs
- If you're uncertain about what a tool returns, acknowledge this
uncertainty
- If multiple tools seemed equally viable or unknown, briefly note why
you selected this one over the others
</tool_preamble>

<quality_checks>
- Review previous tool calls and their results
- Select the strongest available tool for your current needs (or make a
reasonable choice among equals/unknowns)
- Make exactly one tool call per turn
- Use accumulated tool results across turns to formulate your answer
</quality_checks>
```

Figure 10: BrowseComp Domains exploration prompt, part 2. We append the context, including the recent tool-calls and results, to this prompt before receiving the next tool-call.

```
Analyze search tool performance across N question-answering trajectories
 to generate improved tool descriptions that clearly differentiate when
to use each tool.

<input>
- Question-answering trajectories with tool calls and results
- Search results returned by each tool (content may vary by tool)
- Whether the final answer was correct or incorrect
- Current tool descriptions
</input>

<analysis_requirements>
For each tool:
- Identify consistent patterns in the type and quality of results it
returns
- Determine what distinguishes it from other tools
- Provide concrete proof: cite specific trajectories and queries showing
 these patterns
- Focus on situations where this tool performs differently than others

Evaluation notes:
- IMPORTANT: Compare tools relatively, not absolutely
- A tool is effective if it helps the agent reach the correct answer
- Consider both successful and unsuccessful trajectories
- Focus on: result relevance, information completeness, and query type
suitability
- Don't dismiss a tool just because it was used in failed trajectories -
 focus on whether it provided useful information compared to
alternatives
</analysis_requirements>

<output_per_tool>
**Tool: [name]**

Observed patterns: [Key behaviors identified with specific trajectory
evidence]

Distinguishing characteristics: [What makes this tool different from
others, with examples]

Updated description:
[Concise description stating when to use this tool relative to others]

Reasoning: [Justification based on trajectory evidence]
</output_per_tool>

<final_output>
After analyzing all tools, provide a decision framework for selecting
between tools based on the patterns discovered.
</final_output>

Key: Every claim must reference trajectories. Descriptions must be
comparative (tool X better than Y for Z) not absolute.
```

Figure 11: The BrowseComp domains batch analysis reflection prompt. The trajectories are appended to this prompt.

```
You will receive N LLM responses, each analyzing different batches of
question-answering trajectories. Synthesize these into definitive tool
descriptions.

<synthesis_task>
For each tool:
1. Identify patterns that appear across multiple responses
2. Note contradictions between responses
3. Distinguish true patterns from batch-specific noise
4. Look for emergent patterns that no single analysis identified but
become visible when viewing all analyses together
5. Create ONE final description based on the most reliable patterns

Critical:
- A behavior mentioned in only 1-2 responses is likely batch-specific
noise
- Focus on patterns that multiple independent analyses discovered
- Also identify meta-patterns: behaviors that emerge from the collective
 evidence but weren't explicitly stated in any single response
- When responses conflict, examine their evidence strength
- Final descriptions should capture the tool's strengths/weaknesses but
NOT explicitly name other tools
</synthesis_task>

<output_format>
**Tool: [name]**

Synthesis reasoning:
[Explain which patterns were most consistent across analyses, what
emergent patterns were discovered, how conflicts were resolved, and why
certain behaviors were included/excluded in the final description.]

Final description:
[Single definitive description of when to use this tool. Describe its
characteristics and optimal use cases WITHOUT referencing other tools by
 name. Example: "Best for queries requiring recent information or real-
time data. Returns comprehensive results with detailed snippets. May be
less effective for historical or archival content."]
</output_format>
```

Figure 12: The BrowseComp domains consensus merge reflection prompt. The descriptions generated from the previous step are appended to this prompt.

