# OpenReview forum: "OpaqueToolsBench: Learning Nuances of Tool Behavior Through Interaction"
_ICLR.cc/2026/Conference — Submitted to ICLR 2026_

### Official Review · Reviewer_9Zwe · 2025-10-30

**Soundness:** 3
**Presentation:** 3
**Contribution:** 2
**Rating:** 4
**Confidence:** 4

**Summary:**

This paper introduces OPAQUETOOLSBENCH, a benchmark designed to evaluate LLM agents’ ability to learn and adapt to opaque tools, i.e., tools that are underspecified, lack proper documentation, or exhibit non-transparent behaviors. To address the challenge, the authors propose TOOLOBSERVER, a framework that iteratively improves tool documentation through execution feedback (exploration and reflection). Results across all tasks demonstrate the feasibility of learning from execution trajectories to handle real-world, poorly documented tool APIs.

**Strengths:**

1. The paper targets a realistic and underexplored aspect of tool-augmented LLMs: using and improving opaque tools where documentation is minimal or unreliable. This is a valuable shift from previous works that assume perfectly specified APIs.

2. OPAQUETOOLSBENCH is conceptually clean yet diverse, covering structured, unstructured, and sequential tool-use scenarios (function calling, game-playing, and search composition).

**Weaknesses:**

1. TOOLOBSERVER largely reuses ideas from self-reflection and execution-based revision (e.g., Play2Prompt, Reflexion), differing mainly in when reflection occurs (interleaved rather than pre-task).

2. Previous work like [1] has demonstrate the effectiveness of document refinement. What the main difference between this work and [1]? Could the author provide more explanation or comparison in terms of core technique contributions?


3. As for experiment evaluation, other metrics such as the number of reflection iterations, or fine-grained alignment between learned and ground-truth documentation, could be considered for a further validation.

---

### Reference

[1] From Exploration to Mastery: Enabling LLMs to Master Tools via Self-Driven Interactions

**Questions:**

See weakness above.

---

> ### Author Response · Authors · 2025-11-25
>
> Thank you for your review!  We appreciate that you found our focus on opaque tools as a "realistic and underexplored aspect" and that you found OpaqueToolsBench "conceptually clean yet diverse." We address your specific questions and concerns below:
>
> ---
> ### 1. Distinction from Play2Prompt and Reflexion
>
> > TOOLOBSERVER largely reuses ideas from self-reflection and execution-based revision (e.g., Play2Prompt, Reflexion), differing mainly in when reflection occurs (interleaved rather than pre-task).
>
> We emphasize that **OpaqueToolsBench is the primary contribution of this work**, and ToolObserver serves as a strong baseline to validate that the benchmark's challenges are tractable. However, the unique setting necessitates distinct mechanisms; we clarify that the difference is not merely timing, but the **mechanism of exploration** (vs. Play2Prompt [2]) and the **target of optimization** (vs. Reflexion [3]):
> * **Play2Prompt** performs **"isolated unit testing"**, iteratively generating synthetic user queries to test the tool in a vacuum. ToolObserver uses **trajectory-based learning**: it attempts to solve a real task using a sequence of tool calls based on its current best policy. This is critical for OpaqueToolsBench to capture **sequential dependencies** (e.g., Tool B only works after Tool A), which isolated testing cannot discover.
> * **Reflexion** updates **episodic memory** (appending verbal reflections to the agent context), which causes context bloat and limits learning to the specific instance. ToolObserver instead optimizes **semantic memory** (the shared tool documentation). By compressing insights into the global interface, we maintain a stable context window and enable **zero-shot generalization**, allowing future agents to potentially benefit from prior learning without the overhead of accumulated reflections.
>
> Please see **Meta-Response §3** for a more detailed comparison.
>
> ---
>
> ### 2. Comparison with DRAFT (Qu et al., 2024) [1]
>
> > Previous work like [1] has demonstrate the effectiveness of document refinement. What the main difference between this work and [1]? Could the author provide more explanation or comparison in terms of core technique contributions?
>
> We thank the reviewer for highlighting DRAFT. We clarify that DRAFT falls under the "Isolated Unit Testing" paradigm (like Play2Prompt), as it employs an "explorer" to generate synthetic queries to test tools in a vacuum. **ToolObserver differs fundamentally via trajectory-based learning**, offering two critical advantages for this benchmark:
> 1. **Inter-Tool Dynamics**: DRAFT tests tools independently. ToolObserver updates documentation based on goal-oriented trajectories, capturing **sequential dependencies**  (e.g., Tool B relies on Tool A's output). This is essential for the long-horizon tasks in OpaqueToolsBench but impossible to discover via isolated testing.
>
> 2. **Test-Time Efficiency**: DRAFT shares the exhaustive exploration design of Play2Prompt, requiring upfront investment to test the library. ToolObserver is **structurally more efficient**, exploring **only the specific tools required by the user query**. In test-time settings like BFCL, we therefore expect ToolObserver to maintain a similar 3.5–7.5x token efficiency advantage to the one observed against Play2Prompt.
>
> Please see **Meta-Response §3** for a detailed comparison of these paradigms.
>
> ---

---

> ### Author Response · Authors · 2025-11-25
>
> ### 3. Additional Evaluation Metrics
>
> > As for experiment evaluation, other metrics such as the number of reflection iterations, or fine-grained alignment between learned and ground-truth documentation, could be considered for a further validation.
>
> Thank you for this suggestion! We agree that granular metrics provide further insight into the learning process. We **have added the following analyses in our uploaded paper**:
>
> * **Number of Iterations**: We have updated **Section 5.2** (TOOLOBSERVER performance over iterations) and added **Table 5** to report the average iterations to convergence on BFCL (we run a fixed number of improvement iterations for Chess and BrowseComp Domains). The results highlight exceptional sample efficiency: the agent successfully learns opaque tool behaviors in under 3.5 iterations on average.
> * **Documentation Alignment (Semantic Congruence)**: To measure how closely the *content* of our learned understanding matches human intuition, we analyze the similarity between our generated descriptions and the human "Gold" descriptions on BFCL. We focus this analysis on BFCL, where the goal is precise recovery of ground-truth specifications. As shown in **Table 6**, ToolObserver achieves high semantic similarity (**0.78 SBERT**) and lexical overlap (**0.44 ROUGE-1**) in various settings. This confirms that our learned documentation captures both the **semantics and terminology of the ground truth**, validating that ToolObserver effectively reconstructs human-level tool definitions.
> * **Domain-Specific Granular Metrics**: We added Parameter/AST Accuracy (for BFCL) to measure the ability to discover valid input schemas, and Streaming Elo (for Chess) to measure actual gameplay quality. Metric definitions are detailed in **Table 1 and Appendix C**, with full results added to **Tables 2 and 3**. ToolObserver consistently outperforms baselines on these metrics, confirming improvements in both syntactic precision and strategic depth.
>
> ---
>
> ## References
>
> [1] Qu, C., Dai, S., Wei, X., Cai, H., Wang, S., Yin, D., Xu, J., & Wen, J. (2024). From Exploration to Mastery: Enabling LLMs to Master Tools via Self-Driven Interactions. ArXiv, abs/2410.08197.
>
> [2] Fang, Wei-Wen et al. “PLAY2PROMPT: Zero-shot Tool Instruction Optimization for LLM Agents via Tool Play.” Annual Meeting of the Association for Computational Linguistics (2025).
>
> [3] Shinn, N., Cassano, F., Labash, B., Gopinath, A., Narasimhan, K., & Yao, S. (2023). Reflexion: language agents with verbal reinforcement learning. Neural Information Processing Systems.

---

### Official Review · Reviewer_Dj3K · 2025-10-31

**Soundness:** 2
**Presentation:** 3
**Contribution:** 3
**Rating:** 4
**Confidence:** 4

**Summary:**

This work investigates how LLM agents can improve tool use in opaque tool settings by interacting with the tools and iteratively refining their documentation based on execution feedback. The authors propose TOOLOBSERVER, a reflection-based framework that continuously refines tool documentation by observing execution feedback from tool-calling trajectories. In addition, the authors introduce OPAQUETOOLSBENCH, a benchmark for learning in opaque tool settings where tool documentation is underspecified. It consists of three environments: general function calling, interactive chess playing, and long-trajectory agentic search.

Experimental results show that TOOLOBSERVER consistently outperforms all baselines on OPAQUETOOLSBENCH, achieving an average improvement of 18.6% in task success rate while maintaining strong token efficiency.

**Strengths:**

- The paper is well-organized and easy to follow, with clear presentation of benchmark statistics and evaluation metrics.

- The data generation pipeline is simple, scalable, and comprehensively described.

- The simplicity and token efficiency of TOOLOBSERVER make it directly applicable to real-world tool-use scenarios.

**Weaknesses:**

While three domains are tested, the evaluation metrics are somewhat limited. For example, Tables 2 and 3 only report overall accuracy. It would be informative to include additional metrics such as parameter accuracy or Abstract Syntax Tree (measures the generated function call format) etc.

**Questions:**

- In Section 4.3, the authors claim that TOOLOBSERVER is more token-efficient than Play2Prompt. However, this is not fully convincing. Since discovering correct tool usage requires trial and error, calling multiple tools at once may not necessarily reduce overall exploration cost and improve the final accuracy. that is, exploring all tools at once does not sound like a experience-efficiency way to me.

- The opaque tool setting is indeed an underexplored and interesting problem. I wonder whether TOOLOBSERVER could also be applied to refine existing but suboptimal documentation—for instance, improving clarity or usability rather than fixing errors. Similarly, could OPAQUETOOLSBENCH include such cases where the tool documentation is mostly correct but not good enough?

- TOOLOBSERVER explores multiple tools at once, but it remains unclear how the editor generalizes when the number of tools grows substantially. How does performance scale with hundreds or thousands of tools?

- It would be insightful to compare TOOLOBSERVER with human annotators on these tasks. Including a “human oracle” baseline could help quantify how close the model is to human-level understanding of opaque tools.

---

> ### Author Response · Authors · 2025-11-25
>
> Thank you for your review! We appreciate that you found our work "well-organized” and our benchmark as "simple and scalable". We address your specific questions and concerns below:
>
> ---
>
> ### 1) Limited evaluation metrics
>
> > While three domains are tested, the evaluation metrics are somewhat limited. For example, Tables 2 and 3 only report overall accuracy. It would be informative to include additional metrics such as parameter accuracy or Abstract Syntax Tree (measures the generated function call format) etc.
>
> Thank you for this suggestion. **We have added domain-specific metrics** (detailed in **Table 1** and **Appendix C**) that map directly to the two forms of opacity defined in **Meta-Response §2**:
> * **BFCL (Documentation Opacity)**: The core challenge here is discovering valid input schemas from scratch. To measure this **syntactic mastery**, we added **Parameter Accuracy** (exact correctness of arguments) and **AST Accuracy** (syntactical parsability). These metrics reveal that while baselines often fail to structure calls correctly, ToolObserver effectively learns the hidden schema. As shown in **Table 2**, we largely achieve higher parameter recovery, proving the method effectively overcomes documentation opacity.
> * **Chess (Intrinsic Opacity)**: The challenge here is not just selecting a tool, but playing a coherent strategy. To measure this **strategic depth**, we added a **Streaming ELO** metric (**Table 3**) by simulating matches against a diverse pool of three opponents ranging from beginner to master (Stockfish Levels 1,5, 10). We detail the opponent pool and update rules in **Appendix C.3**. While binary accuracy penalizes any deviation from the oracle, ELO captures the *quality* of the resulting gameplay, showing that ToolObserver learns robust winning strategies even when it diverges from the specific engine used by the oracle.
>
> ---
> ### 2) Token efficiency of TOOLOBSERVER vs. Play2Prompt.
>
> > In Section 4.3, the authors claim that TOOLOBSERVER is more token-efficient than Play2Prompt. However, this is not fully convincing. Since discovering correct tool usage requires trial and error, calling multiple tools at once may not necessarily reduce overall exploration cost and improve the final accuracy. that is, exploring all tools at once does not sound like a experience-efficiency way to me.
>
> We find that exploring tools within the actual task trajectory (ToolObserver) is actually **more experience-efficient** than isolated exploration (Play2Prompt) – and often **a necessity for more complex tasks** (e.g., Chess and BrowseComp).
>
> Play2Prompt exhaustively updates the documentation of *every* available tool in a library and and **incurs a heavy upfront cost to generate synthetic scenarios for each one**. In contrast, the task-driven ToolObserver optimizes "on-the-fly" only for the subset of tools retrieved for actual queries, **eliminating this speculative overhead**. In practical "wild" environments with thousands of tools, agents rarely utilize the entire library. In these cases, ToolObserver is significantly more efficient, paying exploration costs **only for the subset of tools relevant to the immediate task**. This targeted approach drives the **3.5x - 7.5x token reduction on BFCL**.
> Finally, we emphasize that **trajectory exploration is essential to capture inter-tool dynamics and sequential dependencies**, which are needed for the complex, longer-horizon tasks (e.g., Chess and BrowseComp). Play2Prompt cannot discover these relationships because its isolated exploration acts as "synthetic unit testing" in a vacuum. This explains our consistent, superior accuracy on these tasks.
>
> We provide a more detailed comparison in **Meta-Response §3**.

---

> > ### Author Response · Authors · 2025-11-25
> >
> > ---
> > ### 3.) Application to suboptimal documentation
> >
> > > The opaque tool setting is indeed an underexplored and interesting problem. I wonder whether TOOLOBSERVER could also be applied to refine existing but suboptimal documentation—for instance, improving clarity or usability rather than fixing errors. Similarly, could OPAQUETOOLSBENCH include such cases where the tool documentation is mostly correct but not good enough?
> >
> > This is a great thought! We find that yes, **TOOLOBSERVER is effective at refining suboptimal documentation**. In fact, **much of OpaqueToolsBench is designed specifically to test this capability** of starting with generic, technically "correct" descriptions and refining them for clarity and usability:
> >
> > 1) In **Chess & BrowseComp** environments, the initial documentation is **accurate but generic** (e.g., "Suggests a chess move" or "Searches the web"). The challenge is not fixing errors, but **improving usability by discovering behavioral nuances** (e.g., "This engine is aggressive in openings but weak in endgames"). TOOLOBSERVER successfully refines these generic descriptions into strategic guides, driving the performance gains in **Tables 3 and 4**.
> >
> > 2) **BFCL-Opaque** includes a scenario specifically for refining existing, high-quality documentation (**Table 2**, 'Anon. Fn. Names + Real Descriptions'). Here, the initial tool descriptions are clear and human-written (e.g., function_1: "Calculates the n numbers of the Fibonacci sequence") but lack structured parameter schemas. To a human, the need for an argument n is semantically obvious here. However with these descriptions, **GPT-5 achieves an initial 0% accuracy overall**. It often correctly identified the right function, but called it without arguments (function_1()) -- a latent model bias that human writers would unlikely predict. ToolObserver refined the documentation to **explicitly enforce the constraint** (e.g., "Call as function_1(n) where n is a positive integer... (e.g., function_1(20))", **recovering performance from 0% to 80%**, and outperforming other baselines.
> >
> > ---
> > ### 4) Scalability with hundreds or thousands of tools.
> >
> > > TOOLOBSERVER explores multiple tools at once, but it remains unclear how the editor generalizes when the number of tools grows substantially. How does performance scale with hundreds or thousands of tools?
> >
> > This is a great point. In principle, ToolObserver can scale to large libraries because it **operates on the retrieved context** (the subset of tools in a specific trajectory) rather than the entire library. **Thus, the optimization cost depends on task complexity (tools per query), not total library size**. However, we acknowledge that performance is eventually bounded by the editor model's reasoning capabilities and context length if a scenario requires coordinating a large number of tools simultaneously within a single trajectory. We view handling such high-concurrency settings as a promising direction for future work.
> >
> > ---
> >
> > ### 5) Comparison with human annotators (Human Oracle)
> >
> > > It would be insightful to compare TOOLOBSERVER with human annotators on these tasks. Including a “human oracle” baseline could help quantify how close the model is to human-level understanding of opaque tools.
> >
> > We agree this is a crucial comparison. We quantify how close our method is to human-level understanding in two ways:
> > 1. Quantifying the Performance Gap (Human Oracle):  We interpret the "Human Oracle" as the **performance achieved using ground-truth, human-authored documentation**. **We add “Gold" baselines to Tables 2 - 4**. The results validate our distinction between opacity types:
> >     * **Syntax Discovery (BFCL)**: ToolObserver **recovers up to 93% of the performance gap** in underspecified settings (0 →0.86 vs. Gold 0.92), demonstrating near-human recovery in some cases. Even in the hardest setting (zero starting information), **it recovers 67% of the gap** (0 → 0.62 vs. 0.92), representing significant progress while highlighting that discovery from no information remains a rigorous challenge.
> >     * **Strategic Reasoning (Chess & BrowseComp)**: A larger gap remains in these domains, confirming that learning complex strategies from scratch is a harder, non-saturated challenge than spec recovery
> >
> > 2. **Quantifying "Human-Level Understanding"**: To measure how closely the *content* of our learned understanding matches human intuition, we analyze the similarity between our generated descriptions and the human "Gold" descriptions on BFCL. We focus this analysis on BFCL, where the goal is precise recovery of ground-truth specifications. As shown in **Table 6**, ToolObserver achieves high semantic similarity (**0.78 SBERT**) and lexical overlap (**0.44 ROUGE-1**) in various settings. This confirms that our learned documentation captures both the **semantics and terminology of the ground truth**, validating that ToolObserver effectively reconstructs human-level tool definitions.

---

### Official Review · Reviewer_mFT3 · 2025-11-01

**Soundness:** 2
**Presentation:** 2
**Contribution:** 2
**Rating:** 4
**Confidence:** 3

**Summary:**

This study designed a situation for tool calling agents when the tool specs are opaque. Under this setting, calling correct tools will be challenging because language models do not have enough information to decide what's the best tool to use or an appropriate parameters to send to the tool.

To evaluate and solve this problem, this study proposed the ToolObserver method that iteratively observe tool behavior and provide incremental improvements to the tool documentations. Experiment shows that the proposed method performs better than selective baselines on tool use benchmarks with opaque tool specs.

**Strengths:**

- The experiment results are encouraging, proving the the proposed method can generate good tool documentations that help improve the performance of models.
- Compared to the baseline method, the proposed strategy does not have to process all tools. as a result, they end up significantly save generation tokens for tool documents.

**Weaknesses:**

I feel the given situation is over-complicated / not well motivated. There are several reasons that tools won't be opaque in most applications:
1. agent developers are trying their best to improve the performance. giving the agent good tool documentation is among the easiest improvement they can do.
2. tool developers would maximize the chance that their tool gets called. as a result, they will work on improving the tool documentation so they are easy for the models to understand.
3. the best use case of the proposed situation might be the agent developers do not know what models they are giving the agent, and the tools developers does not want to tell models what are the tools designed for. this is a very rare case.

Secondly, I think the difference of the proposed strategy of the method and P2P is not strong enough for two reasons:
1. ToolObserve does not need initial documentation, but it does need the schema of tool inputs. as a result, the difference of not needing initial docs is just an incremental steps that tells an LLM to predict tool functionality based on inputs and tool outputs.
2. the token saving mainly comes from that ToolObserver does not have to explore all tools. However, when the number of requests are enough to cover all tools, this claim is no longer valid. ToolObserver and P2P both explore all tools and generates roughly same amount of tokens.

**Questions:**

n/a

---

> ### Author Response · Authors · 2025-11-25
>
> Thank you for your review! We are glad that you found our results "encouraging". We address your specific questions and concerns below:
>
> ---
>
> ### 1. Motivation: Why Opaque Tools are Common and Inevitable
>
> > I feel the given situation is over-complicated / not well motivated. There are several reasons that tools won't be opaque in most applications:
>
> > 1. agent developers are trying their best to improve the performance. giving the agent good tool documentation is among the easiest improvement they can do.
> > 2. tool developers would maximize the chance that their tool gets called. as a result, they will work on improving the tool documentation so they are easy for the models to understand.
> > 3. the best use case of the proposed situation might be the agent developers do not know what models they are giving the agent, and the tools developers does not want to tell models what are the tools designed for. this is a very rare case.
>
> We agree that developers strive for clarity, but OpaqueToolsBench addresses pervasive scenarios where documentation is either practically absent (e.g., legacy enterprise code) or inherently insufficient due to tool complexity (e.g., search ranking logic). Please see **Meta-Response §2** for our full taxonomy of opacity.
>
> Crucially, our results show that **even "good" human documentation often fails models**. This is empirically demonstrated in **BFCL** (**Table 2** under the "Anon. Fn. Names + Real Descriptions" setting). Here, the initial tool descriptions are human-written (e.g., "*function_1: Calculates the n numbers of the Fibonacci sequence*") but lack structured parameter schemas. To a human, the need for an argument *n* is semantically obvious here. However with these descriptions, **GPT-5 achieves an initial 0% accuracy**, often calling the correct function but without arguments (*function_1()*) -- a latent model bias that human writers would unlikely predict. ToolObserver refined the documentation to explicitly enforce the constraint (e.g., "*Call as function_1(n) where n is a positive integer... (e.g., function_1(20))*", **recovering performance from 0% to 80%**. This demonstrates that **tool descriptions often must be optimized for agents beyond human documentation**, to account for the unpredictable ways agents interact with tools.
>
> ---

---

> > ### Author Response · Authors · 2025-11-25
> >
> > ### 2. Differentiation from Play2Prompt (P2P)
> >
> > > Secondly, I think the difference of the proposed strategy of the method and P2P is not strong enough for two reasons:
> >
> > We emphasize that OpaqueToolsBench is the primary contribution of this work. ToolObserver is designed as a strong baseline to demonstrate that the specific challenges of this benchmark (sequential dependencies and state-dependent behaviors) are solvable. However, ToolObserver differs from Play2Prompt in two critical ways necessitated by this new setting (detailed fully in **Meta-Response §3**):
> >
> > **A. Independence from Schemas**
> > > 1. ToolObserve does not need initial documentation, but it does need the schema of tool inputs. as a result, the difference of not needing initial docs is just an incremental steps that tells an LLM to predict tool functionality based on inputs and tool outputs.
> >
> > We would like to clarify that **ToolObserver does not require the schema of tool inputs**. It is capable of discovering valid inputs entirely from scratch by observing execution feedback (e.g., error messages).
> >
> > We validated this in the BFCL "Anon. Fn. Names" setting (**Table 2**), where agents initially received only function names – no descriptions and no parameter inputs. ToolObserver outperformed Play2Prompt across agents while using **~7.5x fewer tokens**. Crucially, we explicitly adapted Play2Prompt for this comparison (as it natively requires schemas), yet it still struggled to synthesize inputs in a vacuum, whereas ToolObserver succeeded by **grounding discovery in the specific task context**.
> >
> > **B. Efficiency and Exploration Quality**
> > > 2. the token saving mainly comes from that ToolObserver does not have to explore all tools. However, when the number of requests are enough to cover all tools, this claim is no longer valid. ToolObserver and P2P both explore all tools and generates roughly same amount of tokens.
> >
> > ToolObserver is **structurally more efficient by design**, even if 100% of tools must be explored. Play2Prompt incurs a heavy upfront cost to **exhaustively generate and validate synthetic scenarios for every tool** in isolation. ToolObserver eliminates this speculative overhead; we drive discovery via the **actual task context**, generating documentation sufficient to solve the task instead of expending tokens on theoretical edge cases.
> >
> > Furthermore, this trajectory-based approach captures **inter-tool dynamics and sequential dependencies** that Play2Prompt misses, explaining our superior accuracy on the longer-horizon tasks (e.g., Chess and BrowseComp).
> >
> > Finally, we note that in practical "wild" environments with thousands of tools, agents rarely utilize the entire library. In these cases, ToolObserver is even more efficient, paying exploration costs **only for the subset of tools relevant to the immediate task**, whereas Play2Prompt demands an exhaustive upfront investment.

---

### Official Review · Reviewer_WVQD · 2025-11-01

**Soundness:** 2
**Presentation:** 3
**Contribution:** 2
**Rating:** 4
**Confidence:** 3

**Summary:**

This paper investigates whether LLM agents can improve their performance when using opaque tools by interacting with them and refining their understanding through feedback. To study this, the authors introduce OPAQUETOOLSBENCH, a benchmark covering three domains: general function calling, interactive chess playing, and long-horizon agentic search. The study finds that existing automatic tool documentation methods are unreliable and costly under opaque conditions. To address this, the authors propose TOOLOBSERVER, a framework that iteratively refines tool documentation based on execution feedback from tool-calling trajectories.

**Strengths:**

1. The concept of opaque tool invocation is novel and opens an interesting direction for further research.
2. The experiments are extensive and well-aligned with the proposed idea.
3. The paper presents a clear analysis, demonstrating how performance evolves across iterations and documentation levels.

**Weaknesses:**

1. The exploration and reflection phases in offline mode lack sufficient detail. The paper provides only a high-level description of these phases without sufficient algorithmic or implementation details.
2. ToolObserver offers limited novelty and resembles prior reflection-based methods.
3. The three benchmark scenarios may not fully represent real-world tool use.
4. Benchmark performance remains low. Even with optimization, the best reported results are still modest, which raises concerns about the practical usefulness and scalability of the proposed method.

**Questions:**

1. The definition of "opague" is relatively obscure. Can you explain it further?
2. If the tasks are not opqgue, how well the models performance?

---

> ### Author Response · Authors · 2025-11-25
>
> Thank you for your review! We appreciate that you found our opaque tool formulation "novel" and our experiments "extensive and well-aligned". We address your specific questions and concerns below:
>
> ---
> ### 1. Lack of detail in Offline Mode exploration/reflection phases.
>
> > The exploration and reflection phases in offline mode lack sufficient detail. The paper provides only a high-level description of these phases without sufficient algorithmic or implementation details.
>
> We have updated **Section 4.1** to provide more clarity in the details of the exploration and reflection phases. We specifically detail the **hierarchical consensus mechanism**, a two-stage process where an "Editor" model first analyzes mini-batches of trajectories to identify causal links (**Batch Analysis**), and then aggregates these insights to filter noise and retain consistent behavioral rules (**Consensus Merge**). Implementation details, including hyperparameters and prompts, are added to **Appendix A and E**.
>
> ---
> ### 2. Novelty and resemblance to prior reflection methods
>
> > ToolObserver offers limited novelty and resembles prior reflection-based methods.
>
> We agree that ToolObserver builds on the foundations of prior reflection-based work. This is intentional, as OpaqueToolsBench is the primary contribution of this work, and ToolObserver serves primarily as a strong baseline to validate it. However, a necessary adaptation was required to handle this new setting. We found that prior methods are ill-suited for OpaqueToolsBench: Play2Prompt [1] relies on "Isolated Unit Testing" which **fails to capture sequential dependencies**, while EasyTool [2] relies on condensing existing text which **fails to learn beyond existing specifications**. ToolObserver adapts reflection to **trajectory-based learning** to demonstrate that these complex, opaque tasks are tractable. We provide a more detailed comparison in **Meta-Response §3**.
>
> ---
> ### 3. Representativeness of benchmark scenarios.
>
> > The three benchmark scenarios may not fully represent real-world tool use.
>
> We designed these environments to isolate the core challenges of opaque tool usage, which are identified in Section 3 (e.g., structured inputs, process feedback, and sequential learning). Each environment serves as a precise proxy for a real-world friction point, mapping to the taxonomy defined in **Meta-Response §2**:
> 1) **BFCL-Opaque** proxies **Documentation Opacity**: This mirrors the common reality of legacy enterprise code or internal tools where specifications are missing or inaccurate.
> 2) **Chess & BrowseComp** proxy **Intrinsic Opacity**: These represent tools with hidden, state-dependent behaviors and inherent complexity – such as complex simulations or LLM-based tools – where behavioral nuances are difficult to fully specify without interaction.
>
> OpaqueToolsBench is the **first benchmark designed to systematically operationalize the opaque tools setting**.
>
> ---
>
> ### 4. Low benchmark performance and 6. Gold oracle baselines
>
> > Benchmark performance remains low. Even with optimization, the best reported results are still modest, which raises concerns about the practical usefulness and scalability of the proposed method.
>
> > If the tasks are not opqgue, how well the models performance?
>
> We view the current performance levels as a confirmation of the benchmark’s design goals rather than a limitation of the method. We explicitly designed OpaqueToolsBench as a rigorous challenge to provide substantial headroom for future research, ensuring the task is not immediately saturated. To quantify this headroom (6.) and demonstrate practicality (4.), we added **Gold Oracle (perfect documentation) baselines to Tables 2–4**:
>
> * **High Recovery on Documentation Opacity (BFCL)**: ToolObserver is effective at recovering missing specifications. In the "underspecified" setting (descriptions provided, but no parameters), we recover **93% of the performance gap** relative to the gold oracle (0.00→0.86 vs. Gold 0.92). Even in the hardest setting (no descriptions or parameters), we recover 67% (0.00→ 0.62 vs. Gold 0.92). This confirms the method is highly effective for "broken" documentation in real-world scenarios.
> * **Rigorous Challenge on Intrinsic Opacity (Chess/BrowseComp)**: A larger gap remains between our method and the Gold Oracle in these domains. This confirms that the benchmark successfully isolates complex, context-dependent capabilities – such as strategic planning or long-horizon search – that are fundamentally harder than spec recovery. The remaining gap validates OpaqueToolsBench as a **rigorous, non-saturated testbed** for future algorithmic improvements.

---

> > ### Author Response · Authors · 2025-11-25
> >
> > ---
> > ### 5. “Opaque” definition
> >
> > > The definition of "opague" is relatively obscure. Can you explain it further?
> >
> > We have updated Section 3 in the paper to explicitly define the two forms of opacity modeled in our benchmark: **documentation opacity** (inaccurate specifications) and **intrinsic opacity** (inherent tool complexity where behavioral nuances are difficult to specify without interaction). We provide a more detailed comparison of these paradigms in **Meta-Response §2**.
> >
> > ---
> >
> > ## References
> >
> > [1] Fang, Wei-Wen et al. “PLAY2PROMPT: Zero-shot Tool Instruction Optimization for LLM Agents via Tool Play.” Annual Meeting of the Association for Computational Linguistics (2025).
> >
> > [2] Yuan, S., Song, K., Chen, J., Tan, X., Shen, Y., Kan, R., Li, D., & Yang, D. (2024). EASYTOOL: Enhancing LLM-based Agents with Concise Tool Instruction. ArXiv, abs/2401.06201.

---

### Author Response · Authors · 2025-11-25
**Meta Response**

We thank the reviewers for their constructive feedback and for recognizing the novelty of our OpaqueToolsBench. Reviewers described the benchmark as "conceptually clean yet diverse" (R-9Zwe), the experiments as "extensive and well-aligned" (R-WVQD), and the problem setting as "realistic and underexplored" (R-9Zwe).

# 1. Common Questions and Responses
## §1: Clarifying the Primary Contribution: OpaqueToolsBench
We emphasize that **OpaqueToolsBench is the primary contribution of this work**. Existing tool-use benchmarks assume near-perfect documentation, which ignores the reality of "wild" tool environments. Our benchmark spans three distinct modalities (structured APIs, strategic game playing, and agentic search) to test diverse forms of opacity. **ToolObserver is introduced as a strong baseline to validate this benchmark**. Its role is to demonstrate that the specific challenges posed by OpaqueToolsBench (e.g., tool sequencing) are *tractable* via trajectory-based interaction, establishing a **rigorous but feasible challenge for the community.**
## §2: Motivation: The Inevitability of Opaque Tools
We clarify why opaque tools are unavoidable in practice. We emphasize that opacity is often structural, not just due to negligence. To formalize this, **we have updated Section 3 to explicitly define the distinct forms of opacity modeled in our benchmark**:
* **Intrinsic Opacity**: Many tools are opaque due to inherent complexity (e.g., search engines, LLM-based tools, or complex simulations). These tools often have a simple schema but complex, undocumented behavioral nuances (e.g., ranking logic or neural edge cases) that even the creator cannot fully capture a priori. The agent must learn these nuances through interaction.
* **Documentation Opacity**: Inaccurate/missing documentation is often unintentional but inevitable in legacy enterprise systems (due to "tribal knowledge" and technical debt) or "wild" marketplaces like the Model Context Protocol (MCP), where diverse authorship leads to inconsistent descriptions at scale.
## §3: Comparison with Prior Work: ToolObserver vs. Play2Prompt and DRAFT
Reviewers noted similarities between ToolObserver and reflection methods like Play2Prompt [1] or DRAFT [2]. However, the unique challenges of OpaqueToolsBench necessitate a distinct approach. **These baselines rely on isolated unit testing** – generating synthetic queries to test every tool in isolation, **failing to capture sequencing or inter-tool dependencies** (e.g., "Tool B works only after Tool A"), which are critical for the longer-horizon tasks in our benchmark. In contrast, ToolObserver utilizes trajectory-based learning, optimizing documentation only as-needed via actual trajectory feedback. This allows it to master the intrinsic opacity of the more complex tasks (Chess/BrowseComp) and makes it significantly more token-efficient (3.5–7.5x) on test-time settings like BFCL, as it avoids both the speculative overhead of generating synthetic scenarios and the cost of optimizing irrelevant tools.

# 2. Updates to the Paper (1/2)
We have uploaded a revised PDF incorporating the following new baselines, metrics, and clarifications:
* **Gold Oracle Baselines**: We added "perfect documentation" baselines for all environments (results updated in **Tables 2-4**). ToolObserver recovers up to 93% of the performance gap on BFCL (validating task feasibility), while larger gaps remain on Chess and BrowseComp (confirming these long-horizon tasks present a rigorous, non-saturated challenge).
* **Granular Metrics**: We added Parameter/AST Accuracy (for BFCL) to measure the ability to discover valid input schemas, and Streaming Elo (for Chess) to measure actual gameplay quality. Metric definitions are detailed in **Table 1 and Appendix C**, with full results added to **Tables 2 and 3**. ToolObserver consistently outperforms baselines on these metrics, confirming improvements in both syntactic precision and strategic depth.
* **Documentation Fidelity**: We added a new quantitative analysis (**Table 6; "Fidelity of learned descriptions”**) measuring the semantic (SBERT) and lexical (ROUGE-1) similarity between the learned tool descriptions and the ground truth on BFCL. We focus this quantitative analysis on BFCL (Documentation Opacity), where the goal is strict recovery of ground truth specifications. Results confirm that our method captures both general semantics and specific terminology significantly better than baselines.
* **Analysis of Learning Dynamics**: We added convergence data for BFCL to **Section 5.2 under “ToolObserver performance over iterations” (Table 5)**. Results show the method is highly sample-efficient, converging in <3.5 iterations on average.

---

> ### Author Response · Authors · 2025-11-25
> **Meta Response Continued**
>
> # 2. Updates to the Paper (2/2)
> * **Method Offline Mode Details**: We expanded the explanation of the offline ToolObserver framework in *Section 4.1*, explicitly detailing the two-stage "Batch Analysis" and "Consensus Merge" reflection process used to filter noise and scale to large datasets. We also updated **Appendix A & E** to provide more implementation details, including hyperparameters and prompts.
> * **Formalized Opaque Tool Taxonomy**: We updated **Section 3** to explicitly define "Intrinsic" vs. "Documentation" opacity. This provides a clearer motivation of the opaque tool settings in our benchmark.
>
> ---
> ### References
> [1] Fang, Wei-Wen et al. “PLAY2PROMPT: Zero-shot Tool Instruction Optimization for LLM Agents via Tool Play.” Annual Meeting of the Association for Computational Linguistics (2025).
>
> [2] Qu, C., Dai, S., Wei, X., Cai, H., Wang, S., Yin, D., Xu, J., & Wen, J. (2024). From Exploration to Mastery: Enabling LLMs to Master Tools via Self-Driven Interactions. ArXiv, abs/2410.08197.

---

### Meta-Review · Area_Chair_LSgN · 2026-01-07

**Summary:**

This paper introduces OpaqueToolsBench, a benchmark for evaluating LLM agents under opaque tool settings, along with ToolObserver for refining tool documentation through execution feedback. Reviewers acknowledged the clean benchmark design but still recommended rejection. The main issues are limited technical innovation compared to existing work, weak motivation for the opaque setting, and concerns about efficiency and performance.

**Reviewer Concerns:**

**Addressed Concerns**

The authors improved the paper during the rebuttal by adding Gold Oracle baselines and more detailed metrics such as AST accuracy. These additions help clarify the experimental results and the gap between the proposed method and ideal conditions.

**Unresolved Concerns**

Despite these improvements, the original concerns raised in the reviews remain:

1. Limited Novelty: The proposed method shares similarities with existing work such as Play2Prompt. The shift from isolated testing to trajectory-based learning is characterized as a relatively minor adaptation rather than a distinct technical contribution.
2. Motivation: The practical necessity of the fully opaque setting remains insufficiently established. The reviews raised concerns that, in many real-world applications, tool documentation is typically provided or iteratively improved, reducing the prevalence of completely undocumented tools.
3. Efficiency: Questions remain regarding the cost of tool exploration during inference and the modest absolute performance observed on more complex benchmark settings.

**Reviewer Scores:**

All four reviewers assigned a score of 4 (marginally below the acceptance threshold). The authors provided a thorough rebuttal, including Gold Oracle baselines, additional granular metrics, and convergence analysis, which strengthened the empirical evaluation. However, these additions did not substantially address the core concerns regarding novelty and problem motivation.

---

### Decision · Program_Chairs · 2026-01-26

Reject